# DESIGN OF LIGAND-BINDING PROTEINS WITH ATOMIC FLOW MATCHING

## ABSTRACT

Designing novel proteins that bind to small molecules is a long-standing challenge in computational biology, with applications in developing catalysts, biosensors, and more. Current computational methods rely on the assumption that the binding pose of the target molecule is known, which is not always feasible, as conformations of novel targets are often unknown and tend to change upon binding. In this work, we formulate proteins and molecules as unified biotokens, and present ATOMFLOW, a novel deep generative model under the flow-matching framework for the design of ligand-binding proteins from the 2D target molecular graph alone. Operating on representative atoms of biotokens, ATOMFLOW captures the flexibility of ligands and generates ligand conformations and protein backbone structures iteratively. We consider the multi-scale nature of biotokens and demonstrate that ATOMFLOW can be effectively trained on a subset of structures from the Protein Data Bank, by matching flow vector field using an SE(3) equivariant structure prediction network. Experimental results show that our method can generate high-fidelity ligand-binding proteins and achieve performance comparable to the state-of-the-art model RFDiffusionAA, while not requiring bound ligand structures. As a general framework, ATOMFLOW holds the potential to be applied to various biomolecule generation tasks in the future.

## 1 INTRODUCTION

Proteins are indispensable macromolecules that drive the essential processes of living organisms. A crucial mechanism by which they accomplish this is through binding with small molecules (Schreier et al., 2009). Continuous progress has been made to design ligand-binding proteins with various biological functions, such as catalysts and biosensors (Bennett et al., 2023). However, the problem remains challenging due to the complex interactions between proteins and molecules, as well as the inherent flexibility of ligands. The most well-established approaches depend on shape complementarity to dock molecules onto native protein scaffold structures (Bick et al., 2017; Polizzi & DeGrado, 2020), which are computationally expensive.

Recently, RFDiffusionAA (Krishna et al., 2024), a de novo protein design method based on the all-atom structure prediction model RoseTTAFoldAA (Krishna et al., 2024), has shown remarkable performance in designing novel ligand-binding proteins for small molecules. This method explicitly captures the interactions between proteins and molecules, achieving superior performance compared to its predecessor RFDiffusion (Watson et al., 2023), which can only model interactions between amino acid residues. Despite their great potential for ligand-binding protein design, current approaches assume that the bound conformation of the target molecule is known and rigid. However, the binding pose of the target molecule is not always available, especially for molecules that do not bind to any known natural proteins (Bick et al., 2017). While it is possible to mitigate this limitation by sampling a diverse set of conformers and subsequently filtering them using expert knowledge (Krishna et al., 2024), this approach demands potentially prohibitive computational resources. Additionally, the constraint of ligand rigidity is suboptimal, as ligands often undergo significant conformation changes upon binding with proteins (Mobley & Dill, 2009). We illustrate this phenomenon in Figure.1. Some pioneering efforts have been made to account for ligand flexibility (Zhang et al., 2024; Stark et al., 2024), however, these methods can only design the portions of proteins that directly interact with the ligands and require the rest part of the proteins as input.

To address the aforementioned issues, we present Atomic Flow-matching (ATOMFLOW), a novel deep generative model grounded in the flow-matching framework (Lipman et al., 2022) for the design of ligand-binding proteins from 2D molecular graphs alone. We model different types of biomolecules within a unified framework that operates in a shared spatial representation, enabling seamless interaction between them, with a flow matching model that directly designs the interactions. Instead of relying on a fixed ligand conformer, ATOMFLOW learns to update the ligand structure along with the structure of the protein binder. Inspired by recent advances

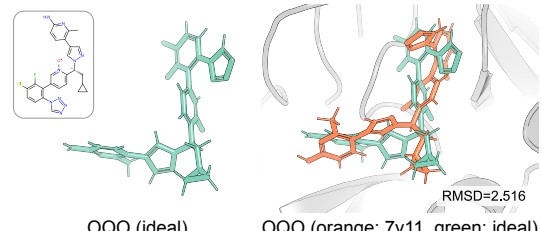

OQO (ideal)   OQO (orange: 7v11, green: ideal)

Figure 1: The conformer of OQO deforms upon binding to coagulation factor XIa. Green: ideal conformer. Orange: bound conformer.

in all-atom structure modeling (Krishna et al., 2024; Abramson et al., 2024), we conceptualize proteins and molecules as biotokens with representative atoms, which are associated with various type-specific attributes and can be modeled by a single, unified network. Following the rectified flow approach (Liu et al., 2022) for generative modelling, we define a flow on the representative atoms as a linear interpolation between the bound protein-ligand complex structures and noisy structures. This unified atomic-level approach maximizes the information aggregation between different molecular types (Bryant et al., 2024) and encourages the model to focus on the key interaction patterns. We demonstrate that, with minor approximations, the vector field of the defined flow can be effectively learned using an SE(3)-equivariant structure prediction module and a variant of Frame Aligned Point Error (FAPE) loss (Jumper et al., 2021) that compensates for the multi-scale nature of their geometric features[1]. After training, protein-ligand complex structures can be sampled from the approximated vector field, which iteratively transforms and refines noisy structures based on 2D molecular graphs. The idea of regressing the vector field using a structure prediction module is also explored in a concurrent work (Jing et al., 2024), but their focus is on protein structure prediction. Notably, as a general generative model operating on biotokens, ATOMFLOW is versatile for different molecular types and has the potential to be applied to various biomolecule generation tasks.

We follow the *in silico* evaluation pipeline of the state-of-the-art method RFDiffusionAA, evaluating ATOMFLOW on several key metrics including self-consistency, binding affinity, diversity, and novelty. ATOMFLOW matches the overall performance of RFDiffusionAA and demonstrates advantages in various situations. An ablation study further highlights that when the bound structure is unknown, ATOMFLOW successfully designs protein binders with high binding affinity, whereas RFDiffusionAA can be constrained by its dependence on a fixed, suboptimal ligand structure.

## 2    RELATED WORK

**Ligand-binding Protein Design.** Traditional approaches to ligand-binding protein design mainly rely on docking molecules onto large sets of shape-complementary protein pockets (Polizzi & DeGrado, 2020; Lu et al., 2024). While the screening process can be accelerated with deep learning models (An et al., 2023), conventional methods are computationally expensive and often depend on domain experts (Bick et al., 2017). Recent advances in deep generative models have paved the way for data-driven approaches, and a variety of models have been proposed to design proteins conditioned on binding targets (Shi et al., 2022; Kong et al., 2023; Watson et al., 2023; Zhang et al., 2024). Focusing on molecule binder design, RFDiffusion (Watson et al., 2023) generates novel proteins from scratch, using a heuristic attractive-repulsive potential to measure shape complementarity. The follow-up work RFDiffusionAA (Krishna et al., 2024) improves the performance by explicitly modeling the interactions between proteins and molecules with an all-atom formulation. These approaches assume binding poses of ligands are known and impose rigidity constraints on ligand structures. Another line of research focuses on designing binding pockets for small molecules (Stark et al., 2024; Zhang et al., 2024). While taking ligand flexibility into consideration, they can only design the portions of proteins that interact with the ligands and require the rest part of the proteins as input. Our model also accounts for the ligand flexibility, but is able to design full ligand-binding proteins from 2D molecular graph alone.

---

[1]The size of a protein is often much larger than that of a molecule. The size disparity should be considered when designing flow-matching models for stable training and inference.

**Protein Generative Model and Structure Prediction.** Recently, various deep generative models for protein generation have emerged (Ingraham et al., 2023; Lin & AlQuraishi, 2023; Yim et al., 2023b;a; Wu et al., 2024; Watson et al., 2023; Krishna et al., 2024). For example, Genie (Lin & AlQuraishi, 2023) introduces a diffusion process defined on C$\alpha$ coordinates of proteins and allows for the incorporation of motif structures as conditions. FrameDiff (Yim et al., 2023b) takes a step further by generating novel protein backbone structures using an SE(3) diffusion process applied to residue frames. Its successor, FrameFlow (Yim et al., 2023a), accelerates the generation process by leveraging the flow-matching framework. However, these approaches are tailored for single-chain protein generation and fall short in modeling multiple biomolecules. In contrast, we treat multiple biomolecules, e.g., proteins and molecules, as biotokens and define a novel flow-matching model on their representative atoms. This allows us to design ligand-binding proteins based solely on molecular graphs, effectively capturing the flexibility of biomolecules and the intricate interactions between them. Our work is also related to approaches that perform protein structure predictions within the all-atom framework, such as RoseTTAFoldAA (Krishna et al., 2024) and AlphaFold 3 (Abramson et al., 2024). These methods tokenize various types of biomolecules into unified tokens, aiming to develop a universal structure prediction model for all molecular types presented in the Protein Data Bank. Our ATOMFLOW adopts the same practice, and we believe this formulation can maximize the information flow between proteins and molecules (Bryant et al., 2024), while our structural modeling on the representative atoms encourages the model to focus on the key patterns of biointeractions.

## 3 PRELIMINARIES

### 3.1 NOTATIONS AND PROBLEM FORMULATION

**Notations.** In this work, a protein-ligand complex is represented as a series of $N$ biotokens $\{a_i \mid a_i = (s_i, x_i), i = 1, 2, \ldots, N\}$, where each token $a_i$ corresponds to either a protein residue or a ligand atom, $s_i$ denotes the token type, and $x_i \in \mathbb{R}^3$ denotes the token position, i.e. the coordinate of its representative atom. Let $\mathcal{S}_{\text{protein}}$ and $\mathcal{S}_{\text{atom}}$ be the set of amino acid types and chemical elements, respectively. For protein residues, $s_i \in \mathcal{S}_{\text{protein}}$, with $x_i$ being the position of the C-$\alpha$ carbon. For ligand atoms, $s_i \in \mathcal{S}_{\text{atom}}$, with $x_i$ being the atomic position. We define the protein token set as $\mathcal{P} = \{a_i \mid s_i \in \mathcal{S}_{\text{protein}}\}$, with $N_p = |\mathcal{P}|$ being the number of protein residues, and the ligand token set as $\mathcal{M} = \{a_i \mid s_i \in \mathcal{S}_{\text{atom}}\}$, with $N_m = |\mathcal{M}|$ representing the number of ligand atoms. In our settings, $N = N_p + N_m$. The biotokens are attributed with token-level features $f^{\text{token}} \in \mathbb{R}^{N \times c_t}$ and pair-level features $f^{\text{pair}} \in \mathbb{R}^{N \times N \times c_p}$, where $c_t$ and $c_p$ denote the feature dimensions.

**Problem Formulation.** Given a ligand molecule represented as a chemical graph $\mathcal{G} = (\mathcal{V}, \mathcal{E})$ and a residue count $N_p$ for the protein binder to be designed, we aim to generate a protein-ligand complex, where a conformer of $\mathcal{G}$ is docked to a protein binder with $N_p$ residues. Specifically, by describing the target protein-ligand complex as a series of biotokens, we generate the token positions $\{x_i\}$, with $\mathbf{x}_m = \{x_i \mid a_i \in \mathcal{M}\}$ being a valid conformer for $\mathcal{G}$, and $\mathbf{x}_p = \{x_i \mid a_i \in \mathcal{P}\}$ being a protein binder with high binding affinity to $\mathbf{x}_m$. Following previous works (Krishna et al., 2024; Yim et al., 2023b), we additionally generate the token frames $\{T_i = (r_i, t_i) \mid a_i \in \mathcal{P}\}$ for protein tokens as described in Appendix A.1, which can be used to recover full backbone coordinates of residues. The design of residue types $\{s_i \mid a_i \in \mathcal{P}\}$ is delegated to an existing reverse folding model (Dauparas et al., 2023).

### 3.2 FLOW MATCHING

Building upon the significant success of diffusion models in various generative tasks, flow matching models (Albergo & Vanden-Eijnden, 2022; Liu et al., 2022) allow for faster and more reliable sampling from a distribution learned from data. The generative process of flow matching models is usually defined by a probability path $p_t(x), t \in [0, 1]$ that gradually transforms from a known noisy distribution $p_0(x) = q(x)$, such as $\mathcal{N}(x|0, I)$ for $x \in \mathbb{R}$, to an approximate data distribution $p_1 \approx p_{\text{data}}(x)$. A vector field $u_t(x)$, which leads to an ODE $\frac{\mathrm{d}\phi_t(\mathbf{x})}{\mathrm{d}t} = u_t(\phi_t(\mathbf{x}))$, is used to generate the probability path via the push-forward equation,

$$p_t = [\phi_t]_* p_0 = p_0(\phi_t^{-1}(x))\det\left[\frac{\partial \phi_t^{-1}}{\partial x}(x)\right], \tag{1}$$

which could be approximated with a trainable network $\hat{v}_t(x; \theta)$.

Due to the complexity of defining an appropriate $p_t$ and $u_t$, we could alternatively define a conditional probability path $p_t(x|x_1)$, which is usually derived through a conditional vector field $u_t(x|x_1)$ for each data point $x_1$ (Lipman et al., 2022). The conditional vector field is then approximated with a trainable network $\hat{v}_t(x; \theta)$. Lipman et al. (2022) has proved that the conditional flow matching loss,

$$\mathcal{L}_{\text{CFM}}(\theta) = \mathbb{E}_{t, p_{\text{data}}(x_1), p_t(x|x_1)} \|\hat{v}_t(x; \theta) - u_t(x|x_1)\|, \tag{2}$$

has identical gradients w.r.t. $\theta$ with $\mathcal{L}_{\text{FM}} = \mathbb{E}_{t, p_{\text{data}}(x)} \|\hat{v}_t(x; \theta) - u_t(x)\|$, which means the model can generate a marginal vector field by simply learning from the $x_1$-conditioned vector fields, without access to $p_t(x)$ and $u_t(x)$. After training, a neural ODE is obtained, ready for sampling from $p_0$ to $p_t$ with an ODE solver (Jardine, 2011).

## 4 METHOD

ATOMFLOW adopts a unified biotoken representation to generate the protein binder and ligand structure by learning the joint distribution of the token positions conditioned on a ligand chemical graph $\mathcal{G}$, $p(\{x_i\}|\mathcal{G})$, from known structures of proteins and protein-ligand complexes. To achieve this, we define a rectified flow on the space of all token positions $\mathbf{x} \in \mathbb{R}^{N \times 3}$, and the corresponding vector field is approximated with an SE(3)-equivariant structure prediction module. The structure predicted at the last generation step is adopted as the final result. In this section, we introduce the flow matching model in Section 4.1, the biotoken feature representation in Section 4.2, the structure prediction module in Section 4.3, and the training and inference procedures in Section 4.4. The overview of our method is illustrated in Figure 2.

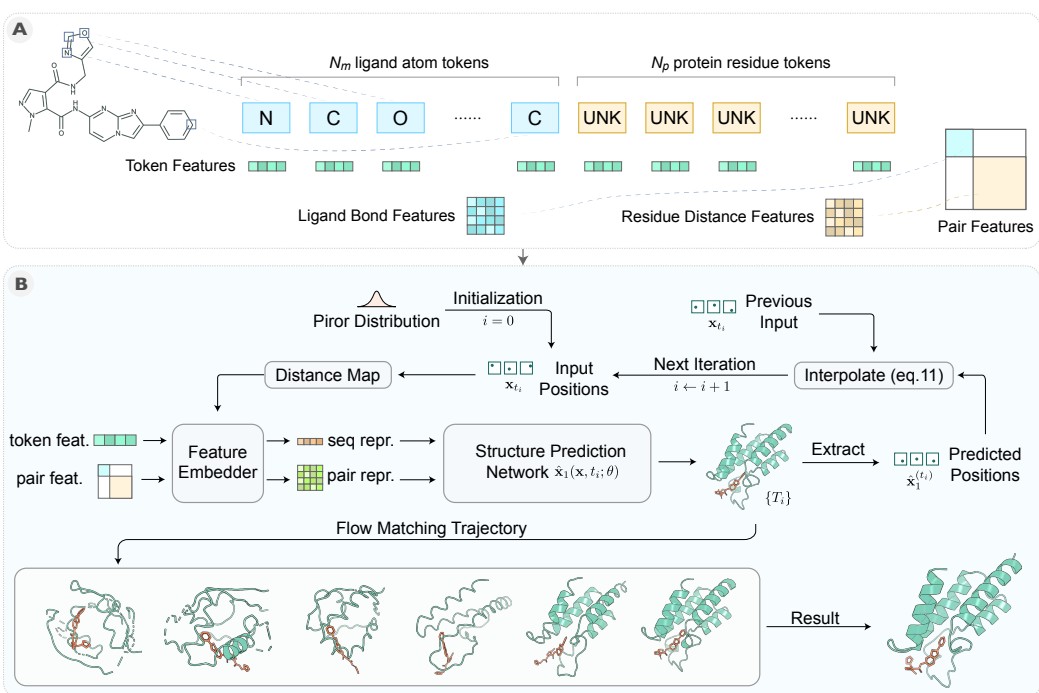

Figure 2: The inference process of ATOMFLOW. We represent the protein-ligand complex as a series of biotokens and embed their token and pair-level features. Starting from a noisy sample, the flow matching procedure gradually generates the designed structure $x_1$ with a structure prediction network.

### 4.1 FLOW MATCHING FOR PROTEIN-LIGAND COMPLEX GENERATION

We jointly design the complex structure $\mathbf{x} = \mathbf{x}_m \cup \mathbf{x}_p$, which lies in the space of $\mathbb{R}^{N \times 3}$, with a flow matching model. Considering that different structures obtained under arbitrary SE(3) transformations correspond to the same complex, we treat each structure as an element in the quotient space $\mathcal{Q}$ : $\mathbb{R}^{N \times 3}/\text{SE}(3)$, where two structures are identical if they could be perfectly aligned with an SE(3)

transformation (Jing et al., 2024). This quotient space is proved to be a Riemannian manifold when defined with suitable care (Diepeveen et al., 2024).

Following Riemannian Flow Matching (Chen & Lipman, 2024), we define a rectified flow on this manifold with a premetric $d : \mathcal{Q} \times \mathcal{Q} \to \mathbb{R}$. We denote $\text{align}_x(y)$ for $x, y \in \mathbb{R}^{N \times 3}$ as aligning structure $y$ to $x$ to minimize RMSD, then the premetric $d(x, y)$ could be defined as the minimum point-wise root mean square deviation (RMSD) among all pairs of possible structures in the original space $\mathbb{R}^{N \times 3}$ for two elements in the quotient space

$$d(x, y) = \min_{\tau \in \text{SE}(3)} \text{RMSD}\left(\tau(y), x\right) = \text{RMSD}\left(\text{align}_x(y) - x\right) \tag{3}$$

**Proposition 1.** *The premetric in equation 3 is a qualified premetric on Q.*

With such premetric at hand, we could obtain a well-defined conditional vector field that decreases the premetric linearly from the prior distribution to the data distribution

$$u_t(x|x_1) = \frac{1}{1-t}\left(\text{align}_x(x_1) - x\right). \tag{4}$$

We leave the proof of Proposition 1 and the derivation of equation 4 to Appendix A.3. Since the vector field is defined as a function of $\mathbf{x}_1$, we could learn the vector field with a structure prediction model $\hat{\mathbf{x}}_1(\mathbf{x}, t; \theta)$. By substituting equations 4 into equation 2, we obtain the training loss

$$\mathcal{L}_{\text{CFM}}(\theta) = \mathbb{E}_{t, p_{\text{data}}(\mathbf{x}_1), p_t(\mathbf{x}|\mathbf{x}_1)} \left\| \frac{1}{1-t}(\text{align}_{\mathbf{x}}(\hat{\mathbf{x}}_1(x, t; \theta)) - \text{align}_{\mathbf{x}}(\mathbf{x}_1)) \right\|, \tag{5}$$

This loss calculates the $(1-t)$-normalized distance between predicted $\hat{\mathbf{x}}_1$ and $\mathbf{x}_1$ in the data distribution aligned to the noisy structure of current step, which is SE(3)-equivariant to both the predicted and ground truth structure. The structure module is designed to predict the token frames (Section 3), while the token positions are extracted from them during the generation process. The last prediction output is adopted as the final result.

Defining a unified flow matching procedure on the joint distribution enables the model to directly learn the structure characteristics that lead to a tightly bound complex, as well as the conformation deformation of both the proteins and the ligands, which is essential to designing a satisfactory ligand-binding protein.

## 4.2 REPRESENTATION OF CONDITIONAL FEATURES

The generation process of ATOMFLOW is conditioned on the ligand chemical graph $\mathcal{G}$ and a designated protein length $N_p$. We model such conditions as an additional condition to the vector field $u$. As a result, the inputs of the prediction network $\hat{\mathbf{x}}_1$ are augmented to accept conditional features. With the biotoken representation, we embed all such features as $f^{\text{token}}$ and $f^{\text{pair}}$ as illustrated in Figure 2A.

For a ligand chemical graph $\mathcal{G}$, we embed the chemical element, as well as other known chemical properties as $f^{\text{token}}$ of ligand tokens. The chemical bonds $\mathcal{E}$ are embedded in $f^{\text{pair}}$ as a multi-dimensional adjacency tensor, each dimension representing a bond type. For residue tokens, we embed the relative residue position (Shaw et al., 2018) in $f^{\text{pair}}$, while $f^{\text{token}}$ may represent other known conditions. We concatenate the protein and ligand features to form a unified feature tensor, eliminating the need to distinguish different types of tokens when processing the features.

## 4.3 STRUCTURE PREDICTION NETWORK

The structure prediction network $\hat{\mathbf{x}}_1(\mathbf{x}, t; \theta)$ [2] predicts the token frames $\{T_i\}$, which can be used to extract token positions $\mathbf{x}_1$, given a series of noisy positions $\mathbf{x}$ at timestamp $t$. It encodes $\mathbf{x}$, along with $f^{\text{token}}$ and $f^{\text{pair}}$, with an SE(3) invariant encoding module, processing the representation with a transformer stack, and generates the predicted structure with a structure module based on invariant-point attention (IPA) (Jumper et al., 2021), as illustrated in Figure 2B. The network jointly processes two kinds of biotokens, protein residues and ligand atoms, with different spatial scales, and handles such differences with special care.

---

[2] Though $\hat{\mathbf{x}}_1$ is a function of $\mathbf{x}, t, f^{\text{token}}, f^{\text{feat}}$, we omitted certain parameters to simplify the text.

**Distance Map.** The input coordinates $\mathbf{x}$ are encoded by projecting the one-hot binned distance map between input coordinates for each token pair to the feature space

$$t_{i,j} = \text{Linear}(\text{BinRepr}(\|\mathbf{x}^{(i)} - \mathbf{x}^{(j)}\|)), \tag{6}$$

where the bins are not divided equally considering the different precision requirements between residues and atoms. This representation is SE(3) invariant, since the internal distance does not change under rigid transformation. [3]

**Feature Embedder.** The feature embedder generates a single representation $s \in \mathbb{R}^{N \times c_s}$ and pair representation $z \in \mathbb{R}^{N \times N \times c_z}$ from distance map $h$, noise level $t$, $f^{\text{token}}$ and $f^{\text{pair}}$ for the following steps. The noise level is encoded with Gaussian Fourier embedding (Song et al., 2021). The local features are concatenated and projected to single representation $s$ and pair representation $z$, $s_i = \text{Linear}(f_i^{\text{local}})$. The pair features and input encoding are projected and added to $z$

$$z_{i,j} = \text{Linear}(f_i^{\text{local}}) + \text{Linear}(f_j^{\text{local}}) + f_{i,j}^{\text{pair}} + t_{i,j}. \tag{7}$$

As described in Section 4.2, different token types can be treated the same and processed uniformly.

**Structure Module.** The structure module generates a predicted complex structure, represented as a series of token frames $T^N$. For ligand atoms, the rotation of the predicted frame is always identity rotation, while the translation equals its position. It first processes $z$ through a deep transformer stack (Appendix A.4) to obtain a denoised pair representation $z'$, and converts $s$ and $z'$ to $T^N$ through a series of shared-weight IPA block

$$T_{1\cdots N} = \text{IPAStack}(s_{1\cdots N}, \text{TransformerStack}(z_{1\cdots N, 1\cdots N})). \tag{8}$$

The IPA stack outputs a sequence of frames for each token, while the rotations for atom tokens are dropped and replaced with the atom frame demonstrated in Section 4.2. The final output represents the full complex structure $\hat{\mathbf{x}}$, while token positions $\hat{x}_1$ are calculated as previously described. The Transformer stack on the unified token sequence allows us to smoothly model the interactions between different types of biological entities in a joint feature space, while the IPA blocks are proved to be efficient when the final structure is properly embedded in the transformer output (Jumper et al., 2021).

**Auxiliary Head.** We add an auxiliary head to predict the pairwise binned distance from the denoised pair representation $z'$, $h_i = \text{softmax}(\text{Linear}(z_i'))$, which directly supervise the input of structure module and has been proved to be helpful during training (Jumper et al., 2021). The bins are also unevenly divided to accommodate the multi-scale characteristics of the predicted complex.

### 4.4 Training and Inference

We train the network $\hat{\mathbf{x}}_1$ by sampling data points and timestamps, calculating the noisy input, and supervising the predicted results. At inference time, we transforms the token positions sampled from the prior distribution through the predicted vector field with an ODE solver, and outputs the structure we obtained at the final step.

**Loss.** We supervise the predicted complex structure $T$ with a metric that measures the structural difference between the observed structure and the predicted structure. Preliminary experiments show that the $\mathcal{L}_{\text{CFM}}$ in equation 5 leads to a fluctuating training trajectory since the aligning object $\mathbf{x}$ varies upon training. With approximation (Appendix A.4), we replace the loss function with a variant of the widely-adopted FAPE function (Jumper et al., 2021),

$$\mathcal{L}_{\text{CFM-FAPE}}(\theta) = \mathbb{E}_{t, p_{\text{data}}(\mathbf{x}_1), p_t(\mathbf{x}|\mathbf{x}_1)} \left[ \frac{1}{1-t} \text{FAPE}(\hat{\mathbf{x}}_1(\mathbf{x}, t; \theta), \mathbf{x}_1) \right]. \tag{9}$$

We show that this substitution does not change the training objective in the appendix. Since the normalization factor $Z$ in the FAPE loss is related to the numerical range of distance, we divide the FAPE loss into protein-protein interaction, protein-ligand interaction, and ligand-ligand interaction, assigning different $Z$s for the three parts. For the auxiliary head, we adopt the cross-entropy loss averaged over all token pairs for the predicted distance. The final training loss

$$\mathcal{L} = \alpha_1 \mathcal{L}_{\text{CFM-FAPE-pp}} + \alpha_2 \mathcal{L}_{\text{CFM-FAPE-pl}} + \alpha_3 \mathcal{L}_{\text{CFM-FAPE-ll}} + \alpha_4 \mathcal{L}_{\text{aux}}. \tag{10}$$

---

[3]To accommodate the precision differences between ligands and proteins, the bin intervals are dense between 1Å (approximate length of a chemical bond) and 3.25Å (approximate distance between adjacent amino acids) and sparser beyond 3.25Å.

Further details are elaborated in Appendix A.4.

**Training.** We sample the timestamp $t$ from the logit normal distribution, assigning more weight on intermediate steps, which helps the model to achieve better performance on hard timestamps (Esser et al., 2024; Karras et al., 2022). The prior distribution $q(x)$ is selected as $\mathcal{N}(0, \sigma_{\text{data}})$, where $\sigma_{\text{data}} = 10$. The input $\mathbf{x}$ is given by interpolating the data point and a sample from the prior distribution. The training procedure is shown in Algorithm 1.

---

**Algorithm 1** Training

**Require:** data distribution $p(\mathbf{x})$, prior distribution $q(\mathbf{x})$, trainable model parameters $\theta$
1: **while** not converged **do**
2:    sample complex structure $\mathbf{x}_1$ and its corresponding ligand chemical graph $\mathcal{G}$ from $p(\mathbf{x}), t \sim [0, 1), \mathbf{x}_0 \sim q(\mathbf{x})$
3:    $N, f^{\text{token}}, f^{\text{pair}} \leftarrow \text{Embedder}(\mathcal{G}, N_p)$
4:    $\mathbf{x}_t \leftarrow t \cdot \mathbf{x}_1 + (1 - t) \cdot \text{align}_{\mathbf{x}}(\mathbf{x}_0)$
5:    $\theta \leftarrow \text{Optimizer}(\theta, (\mathbf{x}_t, f^{\text{token}}, f^{\text{pair}}, t), \mathcal{L})$
6: **end while**
7: **return** $\theta$

---

**Algorithm 2** Inference

**Require:** Chemical graph $\mathcal{G}$, residue count $N_p$, scheduler $t_{0 \cdots m}$, prior distribution $q(\mathbf{x})$, model parameters $\theta$
1: $N, f^{\text{token}}, f^{\text{pair}} \leftarrow \text{Embedder}(\mathcal{G}, N_p)$
2: sample token positions $\mathbf{x}_{t_0} \sim q(\mathbf{x})$
3: **for** $i = 0$ to $m - 1$ **do**
4:    $T_{1 \cdots N} \leftarrow \hat{\mathbf{x}}_1(\mathbf{x}_{t_i}, f^{\text{token}}, f^{\text{pair}}, t_i; \theta)$
5:    $\hat{\mathbf{x}}_1 \leftarrow \text{Extract}(T)$
6:    calculate $\mathbf{x}_{t_{i+1}}$ as Equation 11
7: **end for**
8: **return** $T$

---

**Inference.** A scheduler of noise levels $\{t_i\}_{i=0}^{m}, t_0 = 0, t_m = 1$ is used to determine the noise level $t_i$ of each sampling step $x_{t_i}$. Starting from a noisy sample $x_{t_i} = x_0$ as the initial model input, the structure prediction network predicts the vector field, which gives $x_{t_{i+1}}$ with the Euler's Method, i.e.

$$\mathbf{x}_{t_{i+1}} = \mathbf{x}_{t_i} + \frac{t_{i+1} - t_i}{1 - t_i} \left( \text{align}_{\mathbf{x}_{t_i}} \left( \text{Extract}\big( \hat{\mathbf{x}}_1(\mathbf{x}_{t_i}, t_i; \theta) \big) \right) - \mathbf{x}_{t_i} \right), \quad (11)$$

where the Extract function extracts the token positions from the predicted token frames. The model output at the last step is adopted as the final result. The inference procedure is shown in Algorithm 2.

## 5 Experiments

Following previous protein design models (Yim et al., 2023a; Lin & AlQuraishi, 2023; Watson et al., 2023) and binder design models (Krishna et al., 2024), we evaluate ATOMFLOW through in silico experiments on key metrics of our generated binder including self-consistency, binding affinity, diversity and novelty.

### 5.1 Experiment Setup

**Training Data.** We train the denoising model on two datasets: PDBBind (Liu et al., 2017), a protein-ligand conformer dataset derived from the Protein Data Bank (PDB) (Berman et al., 2000), and SCOPe (Chandonia et al., 2022), a structure categorical dataset for protein. The model is first trained on solely generating the protein structure for 400k steps, and then finetuned on co-generating both the protein and ligand structure for 300k steps.

**Baseline and Model Variant.** We compare ATOMFLOW with the state-of-the-art binder generation method RFDiffusionAA (Krishna et al., 2024), which is extensive trained on almost all known data. Since RFDiffusionAA requires a fixed ligand structure at the binding state as input, we extend our method to work under its setting. For ATOMFLOW, besides the original setting (ATOMFLOW-N), we also train a version of our model with the pairwise distance matrix of the bound structure as an auxiliary hint input (ATOMFLOW-H). This version still needs to generate the ligand structure itself, rather than rely on a fixed structure, as other specifications is not modified. We exclude PocketGen (Zhang et al., 2024) and FlowSite (Stark et al., 2024) since they can only refine the pocket residues of a given binder. We discuss them with an additional experiment in the appendix.

**Evaluation Set.** We mainly evaluate all methods on a selected ligand set (evaluation set) from RFDiffusionAA (FAD, SAM, IAI, OQO). The evaluation set comprises ligands from inside and outside the training set, with both long and short lengths. We conduct evaluations on an extended ligand set (extended set, see Appendix A.5) to further demonstrate the performance of ATOMFLOW.

## 5.2 SELF-CONSISTENCY AND CONFORMER LEGITIMACY

In this section, we evaluate the legitimacy of the generated protein structure by self-consistency RMSD and the predicted ligand structure at the binding state by detecting structural violence in the conformer. Legitimacy is crucial in binder design, given that the model output is not guaranteed to be valid, while a design with higher legitimacy is more likely to fold as expected.

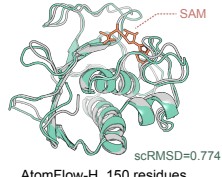 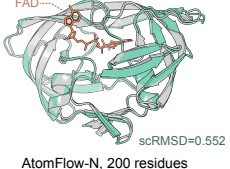 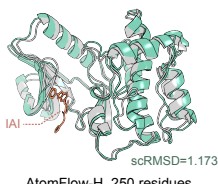 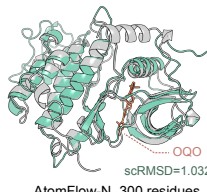

scRMSD=0.774     scRMSD=0.552     scRMSD=1.173     scRMSD=1.032

AtomFlow-H, 150 residues    AtomFlow-N, 200 residues    AtomFlow-H, 250 residues    AtomFlow-N, 300 residues

Figure 3: Designed structures for different ligands at different lengths. We align the ESMFold predicted structure to the designed structure, and report the scRMSD metric. Green: designed protein; Orange: designed ligand conformer; Grey: ESMFold predicted protein.

**Protein Structure.** For protein structures, self-consistency RMSD is widely adopted as a metric to evaluate their legitimacy (Lin & AlQuraishi, 2023; Watson et al., 2023), which compares the generated structure and the folding of its sequence predicted by an accurate model. We adopt LigandMPNN (Dauparas et al., 2023) to predict possible sequences from the generated structures. We first generate 8 sequences for all designed structures with LigandMPNN, then predict the corresponding protein structure with ESMFold (Lin et al., 2023), and the metric for each generated structure is calculated as the minimum rooted mean squared distance between the designed structure and predicted structure (scRMSD). For each ligand in the evaluation set, we generate 10 structures for lengths in [100, 150, 200, 250, 300]. The results are shown in Table 1. We illustrate several generated samples in Figure 3, and the cumulative distribution of scRMSD among them in Figure 4A and 4D. The results on the extended set are shown in Appendix A.5.

| Method | Overall | SAM | FAD | IAI | OQO |
|---|---|---|---|---|---|
| ATOMFLOW-H | **0.57** | **0.60** | 0.36 | **0.58** | **0.74** |
| ATOMFLOW-N | 0.50 | 0.50 | 0.38 | **0.58** | 0.54 |
| RFDiffusionAA | 0.52 | **0.60** | **0.58** | 0.48 | 0.42 |
| RFDiffusion | 0.33 | 0.04 | 0.50 | 0.44 | 0.32 |

Table 1: Proportion of samples with scRMSD $< 2$ on the evaluation set (higher is better).

ATOMFLOW and RFDiffusionAA outperform RFDiffusion on all ligands in the evaluation set, while both ATOMFLOW-H and ATOMFLOW-N reach comparable results to RFDiffusionAA, and exhibit advantages over RFDiffusionAA on several cases. The restricted performance of RFDiffusion is as expected since its binding potential for guiding the protein-ligand interaction may lead to structural destruction. Both ATOMFLOW and RFDiffusionAA model the interaction directly, thus not requiring a strong potential to interfere with the generative process, and leading to better generation results. Notably, without relying on structural guidance from the input ligand conformer, ATOMFLOW-N achieves close performance to ATOMFLOW-H, thereby successfully augmenting the setting to flexible design.

## 5.3 BINDING AFFINITY

In this section, we evaluate the binding affinity of the designed protein binder by calculating an energy function for the atom-level interaction between the protein and the ligand. Binding affinity is the key metric to reveal whether the designed binders are able to bind the target molecule. Though the real binding affinity could only be determined through experiments in the wet lab, an energy function is usually adopted as an in silico alternative (Zhang et al., 2024). We calculate the AutoDock Vina Score (Eberhardt et al., 2021) for all 8 sequences packed by the Rosetta packer (Leaver-Fay et al., 2011), and the reported energy for a structure is the minimum score among all packed proteins. We calculate the energy for all generated structures for the selected ligand set in Section 5.2 and compare ATOMFLOW with RFDiffusionAA and RFDiffusion. The result is illustrated in Figure 4C and 4D.

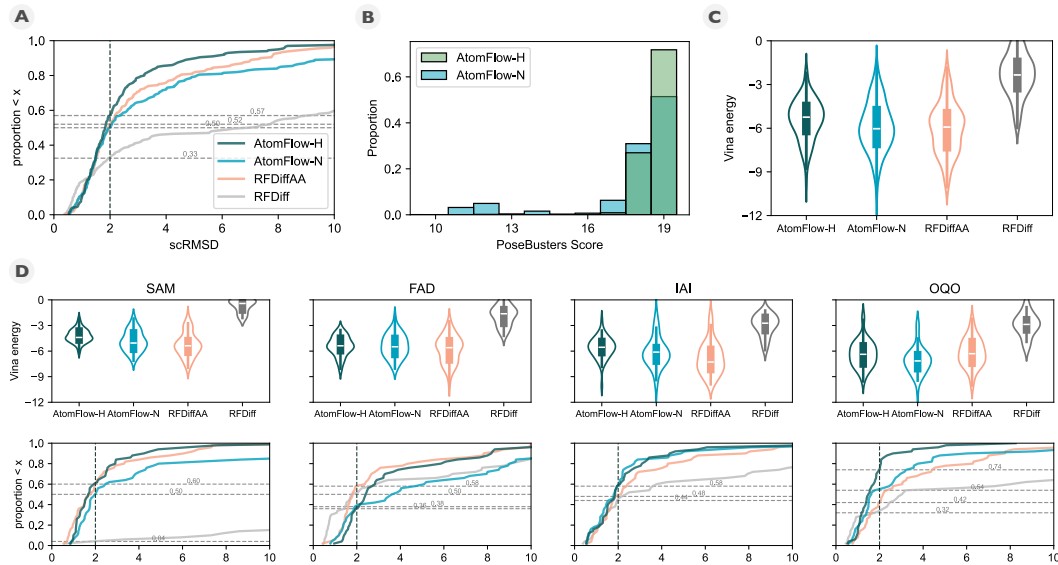

Figure 4: **A**: Self-consistency RMSD distribution curve demonstrating the ratio among all designed samples for the evaluation set with scRMSD $\leq x$ (higher is better). ATOMFLOW outperforms RFDiffusion with a curve similar to RFDiffusionAA. ATOMFLOW-H generates achieves the best result among the methods. The ratio of samples with scRMSD $< 2$ is highlighted. **B**: PoseBusters score distribution of ATOMFLOW generated samples on the extended set. Most ligand conformers generated by ATOMFLOW-N only fail $\leq 1$ metric of its evaluations. **C**: Vina score distribution over all designs on the evaluation set (lower is better). ATOMFLOW achieves comparable performance to RFDiffusionAA, outperforming RFDiffusion. **D**: scRMSD curve and Vina energy distribution over designs for each ligand in the evaluation set. ATOMFLOW outperforms RFDiffusion on all cases and metrics. ATOMFLOW and RFDiffusionAA each exhibit advantages on different ligands, with comparable overall results.

We find that the binding affinity of RFDiffusion is quite poor since it does not model the protein-ligand interaction directly. ATOMFLOW has reached comparable binding affinity to RFDiffusionAA, though marginally lower in several cases. We attribute this to the exhaustive training process of RFDiffusionAA on all known data in PDB, while ATOMFLOW could be further trained and this will be investigated further in our future work. The minimum binding energy generated by ATOMFLOW-H is slightly higher than that of ATOMFLOW-N, possibly because the provided conformer hint hinders the model from exploring additional binding states.

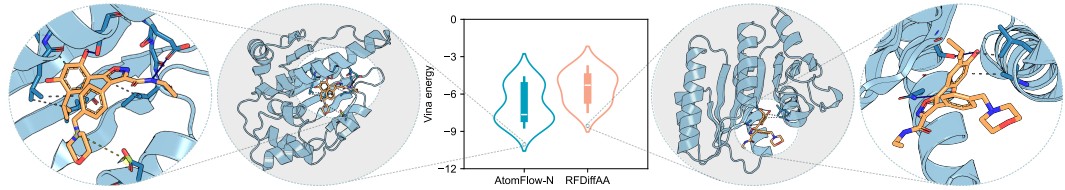

Figure 5: ATOMFLOW-N designs binders with lower vina energy distribution than RFDiffusionAA on 2GJ without the bound structure. Illustration of one sample for each method with PLIP demonstrates that the ATOMFLOW-N designed binder has more chemical interaction with the ligand.

We further compare ATOMFLOW with RFDiffusionAA in a realistic setting where the bound conformer is unknown. We set the target ligand as luminespib (PDB id: 2GJ), an Hsp90 inhibitor (Piotrowska et al., 2018). A designed protein binder for luminespib may act as a protein drug carrier to enhance drug efficacy. Luminespib is a molecule ligand with 33 heavy atoms, so that the conformer is quite flexible when docked to different receptors. We design 10 binders for luminespib using ATOMFLOW and RFDiffusionAA. The ideal conformer from PDB is provided to RFDiffusionAA,

while no conformer is provided to ATOMFLOW. The binding energy of the designed structures and one designed sample with PLIP (Adasme et al., 2021) to demonstrate the protein-ligand interaction are illustrated in Figure 5. It is shown that ATOMFLOW generates more binders with higher binding affinity than RFDiffusionAA, and significantly outperforms RFDiffusionAA on the lowest energy among all generated structures. This demonstrates that a proper bound structure is crucial to the performance of RFDiffusionAA, while ATOMFLOW does not rely on such structure and generates proper conformers by co-modeling the structure space of proteins and ligands.

## 5.4 DIVERSITY AND NOVELTY

In this section, we report the diversity and novelty of ATOMFLOW, following common practice in literature (Krishna et al., 2024; Yim et al., 2023b). Diversity refers to the structural divergence of the designed binders for a certain ligand, while novelty refers to how close a designed protein is to the known proteins. For diversity, we generate 100 structures with 200 residues for each ligand, and then use MaxCluster (Herbert, 2008) to calculate the pairwise structural distance of the outputs and report the number of clusters using different thresholds of maximum distance within a cluster. For novelty, we generate 4 structures with residue count in $[100, 101, \cdots, 300]$ for each ligand, and then calculate the highest TM-score (Zhang, 2005) between a designed structure and any similar structure searched by FoldSeek (Kempen et al., 2024) (pdbTM), as well as the protein scRMSD. The search range of pdbTM is all known protein structures in PDB.

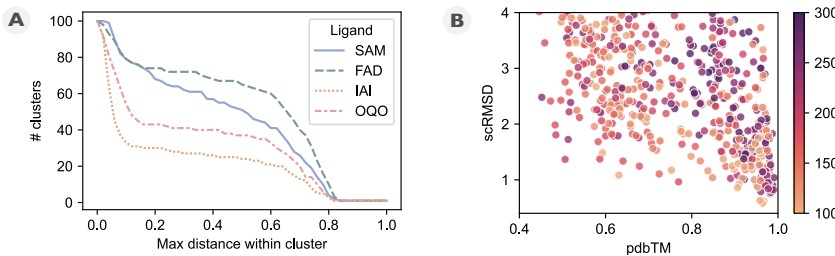

Figure 6: **A**: Cluster count based on different thresholds for the maximum difference within the cluster for each ligand in the evaluation set. ATOMFLOW generates diverse binder folds for all ligands, not restricted to the existing binder structure. **B**: Scatter plot of designability (scRMSD) vs. novelty (pdbTM) for ligands in the evaluation set. ATOMFLOW successfully designs self-consistent structures with high pdbTM, demonstrating high novelty.

Figure 6A shows that the structures generated by ATOMFLOW are quite diverse for all four ligands, and the diversity varies among different ligands. Though existing protein-ligand complexes only provide limited folds for possible binders, by adding protein-only data to the training set, our model successfully learns from the protein structure distribution to generate more possible folds, instead of replicating known patterns. The scatter plot of scRMSD vs. pdbTM shown in Figure 6B reveals that ATOMFLOW has the ability to generate structures that are quite different from existing proteins with acceptable designability. Note that most designable structures are still similar to known ones, which is as expected since most protein folds are already discovered, while novel folds are quite sparse and hard to derive.

## 6 CONCLUSION AND FUTURE WORK

In this work, we proposed ATOMFLOW, a de novo protein binder design method for small molecule ligands considering the flexibility of ligand structure. Unlike previous works, ATOMFLOW no longer relies on a given bound ligand conformer as input. We represent the protein-ligand complex as unified biotokens, learning the structure distribution of both the proteins and the ligands simultaneously from the data with an SE(3)-equivariant flow matching model on the representative atoms. During the evaluation, ATOMFLOW shows comparable design quality to the state-of-the-art model RFDiffusionAA, which requires the ligand conformer to be fixed before design. Further evaluation exhibits the advantage of ATOMFLOW in the circumstance when the ligand conformer is not known. A direct future work is to support more precise control of the generated structures, and we're working to migrate ATOMFLOW to all kinds of biomolecules, including DNA, RNA, and metal ions.

## REPRODUCIBILITY STATEMENT

We include the source code of the AtomFlow model and its corresponding checkpoint with a ready-to-use Gradio (Abid et al., 2019) interface in the supplementary materials. Instructions for setting up the environment and launching the web-based interface are provided as a `README` file. Further details on the model implementation and training are available in Appendix A.4.

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

## A  Appendix

### A.1  Protein Frames

Proteins are composed of amino acid chains linked by peptide bonds, forming a backbone with protruding side chains. Each amino acid's position and orientation are described by a local coordinate system, or protein frame, centered on three key backbone atoms: the alpha carbon (C$\alpha$), the carbonyl carbon (C), and the amide nitrogen (N). These atoms act as reference points for establishing the frame. The alpha carbon (C$\alpha$) typically acts as the origin. The vector from C$\alpha$ to the amide nitrogen (N) is normalized to define one axis of the frame. A second axis is defined by the normalized vector from C$\alpha$ to the carbonyl carbon (C). The third axis is formed by the cross product of these two vectors, creating an orthogonal, right-handed coordinate system. The residue frame is typically represented as an SE(3) transformation $T = (R, t)$, which maps a vector from this local system to the global coordinate system. In this transformation, $t$ corresponds to the position of C$\alpha$ in the global system, and $R$ represents the rotation needed to align the residue's structure within the global context.

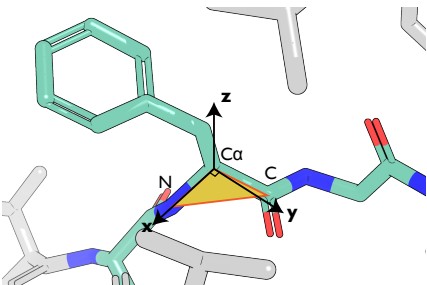

Figure 7: A protein frame illustration. The C$\alpha$, C, N atoms form a panel, which is the xy panel. The x-axis is defined as the orientation from C$\alpha$ to N, while the y-axis is on the panel and perpendicular to the x-axis. The z-axis is perpendicular to the xy panel.

### A.2  Details on Biotokens

**Token Features**  For ligand atom tokens, the token-level feature set includes: chirality, degree, formal charge, implicit valence, number of H atoms, number of radical electrons, orbital hybridization, aromaticity, and ring size. The pair-level feature is provided as one-hot embedding of the bond type. For residue tokens, no token-level feature is known, while the pair-level feature only contains the binned distance of residue index between residues. All features are encoded as a one-hot vector and concatenated.

**Token Frames**  The final loss we adopted $\mathcal{L}_{\text{CFM-FAPE}}$ requires aligning the predicted structure to the local frame of every token. The frames of protein residues can be naturally defined as in Section 3. However, the frames of ligand atoms could not be chosen directly. Since a frame could be calculated from the coordinate of 3 atoms, we need to choose an atom triplet for every atom token.

We first obtain a canonical rank of every atom that does not depend on the input order (Schneider et al., 2015). The atoms are then renamed to their rank. For atoms $x$ with a degree greater than or equal to 2, we select the lexicographically smallest triplet $(u, x, v)$ to define the frame, where $u$ and $v$ are neighbors of $x$. For atoms with a degree of 1, $u$ is the only neighbor of $x$, and $v$ is chosen as one of $u$ 's neighbors. This method ensures that each atom's frame is defined in a consistent manner, irrespective of its position in the input sequence, thereby facilitating the model to learn a consistent structural target.

**Extending Token Types and Features**  Though ATOMFLOW only considers the interaction between protein and molecule ligands, the unified biotoken has the potential to extend to all biological entities, including DNA, RNA, etc, by defining the token position, token frame, local and pair features, and the representation of the internal structure. For example, an RNA can be represented as a sequence of nucleotides, with the token position defined as its mass center, and the token frame calculated from an atom triplet, such as C2-N1-C6.

The token features can also be extended to support more types of known information. For example, the local features could also contain an embedding to indicate the preferred secondary structure, or whether a ligand atom is required to be closer to the designed protein; the pair features could also contain the motif information with a distance map.

### A.3 DETAILS ON THE FLOW MATCHING PROCESS

For all types of tokens, we only consider their token positions to simplify the flow matching process. Thus, the positions of all tokens lie in the Euclidean space $\mathbb{R}^{N \times 3}$. Since a complex could be arbitrarily moved or rotated in the coordinate space without changing its structure, we need an algorithm that treats different position series as the same if they could be aligned with an SE(3) translation. Thus, every data point we consider now lies in the quotient space $\mathbb{R}^{N \times 3}/\text{SE}(3)$. This quotient space is proved to be a Riemannian manifold (Diepeveen et al., 2024).

For a Riemannian manifold, the flow matching process could be defined using a premetric (Chen & Lipman, 2024). A premetric $d : \mathcal{M} \times \mathcal{M} \to \mathbb{R}$ should satisfy: 1. $d(x, y) \geq 0$ for all $x, y \in \mathcal{M}$; 2. $d(x, y) = 0$ iff $x = y$; 3. $\nabla d(x, y) \neq 0$ iff $x \neq y$.

We define our premetric as the minimum point-wise rooted sum of squared distance (RMSD) among all pairs of possible structures in the original space $\mathbb{R}^{N \times 3}$ for two elements in the quotient space $d(x, y) = \|\text{align}_x(y) - x\|$, which satisfies all three conditions (Proposition 1).

*Proof.* Since the premetric is defined as a norm, it satisfies condition 1 by nature. When $x = y$, the best alignment that aligns $y$ to $x$ could derive the exact same position as $x$, yielding a zero norm. When $x \neq y$, when $y$ is aligned to $x$, there's still a structural difference between the structures, thus the premetric is not zero. For condition 3, by defining $y' = \text{align}_x(y)$, we have

$$\nabla d(x, y) = \nabla \sqrt{\sum_{i=1}^{n} (y_i' - x_i)^2} = \frac{y' - x}{\|y' - x\|} = \frac{\text{align}_x(y) - x}{\|\text{align}_x(y) - x\|} \geq 0. \tag{12}$$

Thus $d(x, y)$ satisfies all the conditions as a qualified premetric. $\qquad\square$

With such premetric, and a monotonically decreasing differentiable scheduler $\kappa(t) = 1 - t$, we could obtain a well-defined conditional vector field that linearly interpolates between the noisy and real data (Chen & Lipman, 2024)

$$u_t(x|x_1) = \frac{d \log \kappa(t)}{dt} d(x, x_1) \frac{\nabla d(x, x_1)}{\|\nabla d(x, x_1)\|^2} = \frac{1}{1 - t}(\text{align}_x(x_1) - x). \tag{13}$$

The vector field in equation 13 is calculated by substituting equation 12 into the left side. This vector field provides the direction for moving straight towards $x_1$, and generates a probability flow that interpolates linearly between noisy sample $x_0$ and data sample $x_1$.

Since the vector field is defined as a function of $x_1$, we could learn the vector field with a structure prediction model $\hat{x}_1(x, t; \theta)$. By substituting equation 4 into equation 2, we obtain the training loss

$$\mathcal{L}_{\text{CFM}}(\theta) = \mathbb{E}_{t, p_{\text{data}}(x_1), p_t(x|x_1)} \left\| \frac{1}{1 - t}(\text{align}_x(\hat{x}_1(x, t; \theta)) - \text{align}_x(x_1)) \right\|. \tag{14}$$

### A.4 DETAILS ON THE PREDICTION NETWORK

**Structure Module Specifications** The main components of the structure module are derived from Alphafold 2 (Jumper et al., 2021), while our implementation builds on top of the widely acknowledged reimplementation OpenFold (Ahdritz et al., 2024). The TransformerStack consists of 14 layers of simplified Evoformer block, and the IPAStack consists of 4 layers of Invariant Point Attention (IPA) blocks. The MSA operations in the Evoformer block are simplified by replacing the operations on the MSA feature matrix with the single representation $s_i$. The weights of the IPA blocks are shared, and the structural loss is calculated on the outputs of each block and averaged.

**Training Details**    During training, we equally sample data from the SCOPe dataset (v2.08) and the PDBBind dataset (2020). We simply drop the data with more than 512 tokens, and we don't crop the filtered complexes since the cutoff is large enough and only filters out a relatively small portion of the data. We train our model on 10 NIVIDA RTX 4090 acceleration cards, with a batch size set to 10, which means the batch size on each device is set to 1. We use the Adam Optimizer (Kingma, 2014) with a weight-decaying learning rate scheduler, starting from $10^{-3}$ and decays the learning rate by 0.95 every 50k steps. We separate the training process into two stages: 1) initial training, $\alpha_1 = 0.5, \alpha_4 = 0.3, \alpha_2 = \alpha_3 = 0$; 2) finetuning, $\alpha_1 = \alpha_2 = \alpha_3 = 0.5, \alpha_4 = 0.3$.

Ligand tokens are not given during the first training stage. The first stage trains an unconditional protein generation model, while the second stage turns it to a conditional protein binder and ligand conformer generation model. The FAPE loss is defined as an average of all pairs of tokens in the original paper, so the calculation process first yield a FAPE matrix and then produce the average value of the matrix. The protein-protein, protein-ligand and ligand-ligand loss calculates the average value of the sub-matrixs defined as (row: protein, col: protein), (row: protein, col: ligand), and (row: ligand, col:ligand).

Since training a protein design model is significantly time-consuming, the design choices of our training strategy is largely determined by grid searching possible design space and save the training trajectory of the first 30~50k steps. We compare the training trajectories and select the best configuration that meets the following criteria: a) The final distogram loss should close the minimum we get among the configurations (around 2.0). b) The $\mathcal{L}_{\text{CFM-FAPE}}$ should not decline too fast at the first 10k steps. The first 10k steps is for the transformer stack to learn a relatively steady output, indicated by the decline of the distogram loss. A decline of $\mathcal{L}_{\text{CFM-FAPE}}$ at this stage will resulted in an undesired local minimum. Then $\mathcal{L}_{\text{CFM-FAPE}}$ should decline fast right after the distogram loss turns to decline much smoother. We select the configuration with the lowest $\mathcal{L}_{\text{CFM-FAPE}}$ at the end of training.

We decide the end of each training stage when the training converges, with the following criteria: a) the decline rate of every single loss is small. b) the structural violence of sampled structures (counts of CA atom violation) converges.

An initial study on directly train the second stage shows unsatisfactory training trajectory. Since the ligand conformer is way easier to generate compared to protein folds, the FAPE loss declines too fast even before the distogram loss, resulted in unstable TransformerStack output, and leading to a diverge of the model after  30k steps. The resulted model with minimum loss is able to predict the ligand structure, with random protein residue position, which is unusable.

**Loss Function**    $\mathcal{L}_{\text{CFM}}$ calculates an aligned RMSD by aligning $\mathbf{x_1}$ and $\hat{\mathbf{x_1}}$ to $\mathbf{x}$, while the FAPE loss calculates an averaged RMSD by aligning $\hat{\mathbf{x_1}}$ to each residue frame of $\mathbf{x_1}$, which could be extended to the token frame (Appendix A.2). Let $\text{align}_{x,i}(y)$ denote aligning $y$ to the $i$-th token frame of $x$, we have

$$\mathcal{L}_{\text{CFM}} = \mathbb{E}_{t, p_{\text{data}}(x_1), p_t(x|x_1)} \left\| \frac{1}{1-t} (\text{align}_x(\hat{x_1}(x,t;\theta)) - \text{align}_x(x_1)) \right\|$$

$$\approx \mathbb{E}_{t, p_{\text{data}}(x_1), p_t(x|x_1)} \left\| \frac{1}{1-t} \cdot \frac{1}{N} \sum_{i=1}^{N} (\text{align}_{x,i}(\hat{x_1}(x,t;\theta)) - \text{align}_{x,i}(x_1)) \right\|$$

$$\approx \mathbb{E}_{t, p_{\text{data}}(x_1), p_t(x|x_1)} \left\| \frac{1}{1-t} \cdot \frac{1}{N} \sum_{i=1}^{N} (\text{align}_{x_1,i}(\hat{x_1}(x,t;\theta)) - \text{align}_{x_1,i}(x_1)) \right\|$$

$$\approx \mathbb{E}_{t, p_{\text{data}}(x_1), p_t(x|x_1)} \left\| \frac{1}{1-t} \cdot \frac{1}{N} \sum_{i=1}^{N} (\text{align}_{x_1,i}(\hat{x_1}(x,t;\theta)) - x_1) \right\|$$

$$= \mathcal{L}_{\text{CFM-FAPE}}$$

**Proposition 2.** $\text{align}_{\mathbf{x_1}}(\hat{\mathbf{x_1}}) = \mathbf{x_1} \iff \mathcal{L}_{\text{CFM}} = 0 \iff \mathcal{L}_{\text{CFM-FAPE}} = 0.$

*Proof.* When $\text{align}_{\mathbf{x_1}}(\hat{\mathbf{x_1}}) = \mathbf{x_1}$, we have $\forall i, \text{align}_{\mathbf{x_1},i}(\hat{\mathbf{x_1}}) = \mathbf{x_1}$. As a result, $\mathcal{L}_{\text{CFM}} = \mathcal{L}_{\text{CFM-FAPE}} = 0$. This establishes that:

$$\text{align}_{\mathbf{x_1}}(\hat{\mathbf{x_1}}) = \mathbf{x_1} \iff \mathcal{L}_{\text{CFM}} = 0 \quad \text{and} \quad \text{align}_{\mathbf{x_1}}(\hat{\mathbf{x_1}}) = \mathbf{x_1} \iff \mathcal{L}_{\text{CFM-FAPE}} = 0. \tag{15}$$

Now, assume $\mathcal{L}_{\text{CFM}} = 0$. Suppose $\text{align}_{\mathbf{x_1}}(\hat{\mathbf{x_1}}) \neq \mathbf{x_1}$. Then for all transformations $R$ and $t$, we have $R\hat{\mathbf{x_1}} + t \neq \mathbf{x_1}$, which implies: $\|\text{align}_{\mathbf{x_1}}(\hat{\mathbf{x_1}}) - \mathbf{x_1}\| \neq 0$, leading to $\mathcal{L}_{\text{CFM}} \neq 0$. This is a contradiction. Therefore, $\text{align}_{\mathbf{x_1}}(\hat{\mathbf{x_1}}) = \mathbf{x_1}$. This proves that

$$\mathcal{L}_{\text{CFM}} = 0 \iff \text{align}_{\mathbf{x_1}}(\hat{\mathbf{x_1}}) = \mathbf{x_1}. \tag{16}$$

Similarly, assume $\mathcal{L}_{\text{CFM-FAPE}} = 0$. Suppose $\text{align}_{\mathbf{x_1}}(\hat{\mathbf{x_1}}) \neq \mathbf{x_1}$. Then: $\|\text{align}_{\mathbf{x_1},i}(\hat{\mathbf{x_1}}) - \mathbf{x_1}\| \neq 0$, which leads to $\mathcal{L}_{\text{CFM-FAPE}} \neq 0$, again a contradiction. Therefore, $\text{align}_{\mathbf{x_1}}(\hat{\mathbf{x_1}}) = \mathbf{x_1}$. This proves that:

$$\mathcal{L}_{\text{CFM-FAPE}} = 0 \iff \text{align}_{\mathbf{x_1}}(\hat{\mathbf{x_1}}) = \mathbf{x_1}. \tag{17}$$

The proposition is proved by combining equation 15,16,17. □

This means that both $\mathcal{L}_{\text{CFM}}$ and $\mathcal{L}_{\text{CFM-FAPE}}$ provide an optimization direction towards minimizing the SE(3) invariant structural difference between the predicted structure and the ground truth structure. Thus, we adopt $\mathcal{L}_{\text{CFM-FAPE}}$ as a realistic approximation of $\mathcal{L}_{\text{CFM}}$ and adopt it as the training objection during evaluation.

### A.5 EVALUATION DETAILS

**Specifications**  Following RFDiffusionAA, we use FAD, SAM, IAI, and OQO as the selected evaluation set. FAD and SAM are witnessed by both models as training data, while IAI and OQO are not, demonstrating the generalization ability. To further investigate the performance of our method, we conduct experiments on an extended set of 20 ligands (ligands from PDB id 6cjs, 6e4c, 6gj6, 5zk7, 6qto, 6i78, 6ggd, 6cjj, 6i67, 6iby, 6nw3, 6o5g, 6hlb, 6efk, 6gga, 6mhd, 6i8m, 6s56, 6tel, and 6ffe). The extended dataset includes ligand sizes (including hydrogen) ranging from 21 to 104 in length.

**Extended Set**  We illustrate the designability (scRMSD) and binding affinity (Vina energy) of ATOMFLOW-N in Figure 8. The extended evaluation shows that the performance of ATOMFLOW on the extended set is similar to the evaluation set shown in the main article, and demonstrates that ATOMFLOW is able to tackle almost all kinds of ligands.

**Additional Binding Affinity Metric**  We are aware that Vina might not be a perfect proxy for binding affinity. We noticed that AlphaProteo adopts several metrics produced by AlphaFold 3 as in silico filters. At the time we finished our draft, there's no publicly available AlphaFold 3 for us to run locally. Recently, several AlphaFold 3 replications and the original AlphaFold 3 are released. We developed an alternative in silico metric based on Chai-1, calculating the minimum value across all interchain terms in the PAE matrix (`min_ipAE`, lower is better). Note that this metric is proved to be a good indicator for protein-protein binder design, but not validated on small molecule-protein binders. The results in Figure 9 show that the AtomFlow-generated binders have lower `min_ipAE` than the ones from RFDiffusion-AA, and both models have the ability to generate binders with similar `min_ipAE` as natural ones. We're working to develop better metrics as in silico proxy for ligand-binding protein design based on wet lab verification.

**Diversity and Novelty Results of the Baseline.**  We conducted the diversity and novelty experiment on RFDiffusion-AA with the same configuration as our results reported in the main text. The results are shown in Figure 10. The diversity of AtomFlow designs is better than RFDiffusion-AA, while the AtomFlow generated results tends to be more conservative in terms of pdbTM novelty. We believe this is because we didn't train AtomFlow on a full training set including all PDB structures and the distillation data. This is our future work and we'll release an updated model once available.

**Discussion on Pocket Design Models**  While the pocket design models address ligand-protein interactions, their focus is limited to refining pocket residues within a predefined radius. They lack the capacity to design full protein folds, making direct comparison with AtomFlow infeasible. We conducted an unfair experiment with PocketGen by providing a template binder to it, as detailed in Tabel 2. Despite this, the results demonstrate that AtomFlow consistently outperforms PocketGen in terms of fold quality across all radii.

**Geometrical Distributions of Generated Structure** We evaluated the common chemical bond length generated by AtomFlow vs. the ground truth bond length in our training set. Results shown in Figure 11 demonstrate that the AtomFlow generated ligands have similar geometric distribution to ground truth. We further evaluated the generated structures by plotting the Ramachandran plots. Results shown in Figure 12 suggests that the proteins generated by AtomFlow effectively capture the key structural characteristics of natural proteins.

**Chemical Validity** We evaluated the generated ligands with several important chemical validity metrics: QED, an index of drug-likeness, with a value between 0 (drug-unlike) and 1 (drug-like); SA, the difficulty of chemical synthesis for molecules, with a value between 0 (easy to synthesize) and 10 (very difficult to synthesize); LogP, an important parameter to characterize the overall hydrophobicity of organic compounds. Results are shown in Tabel 3.

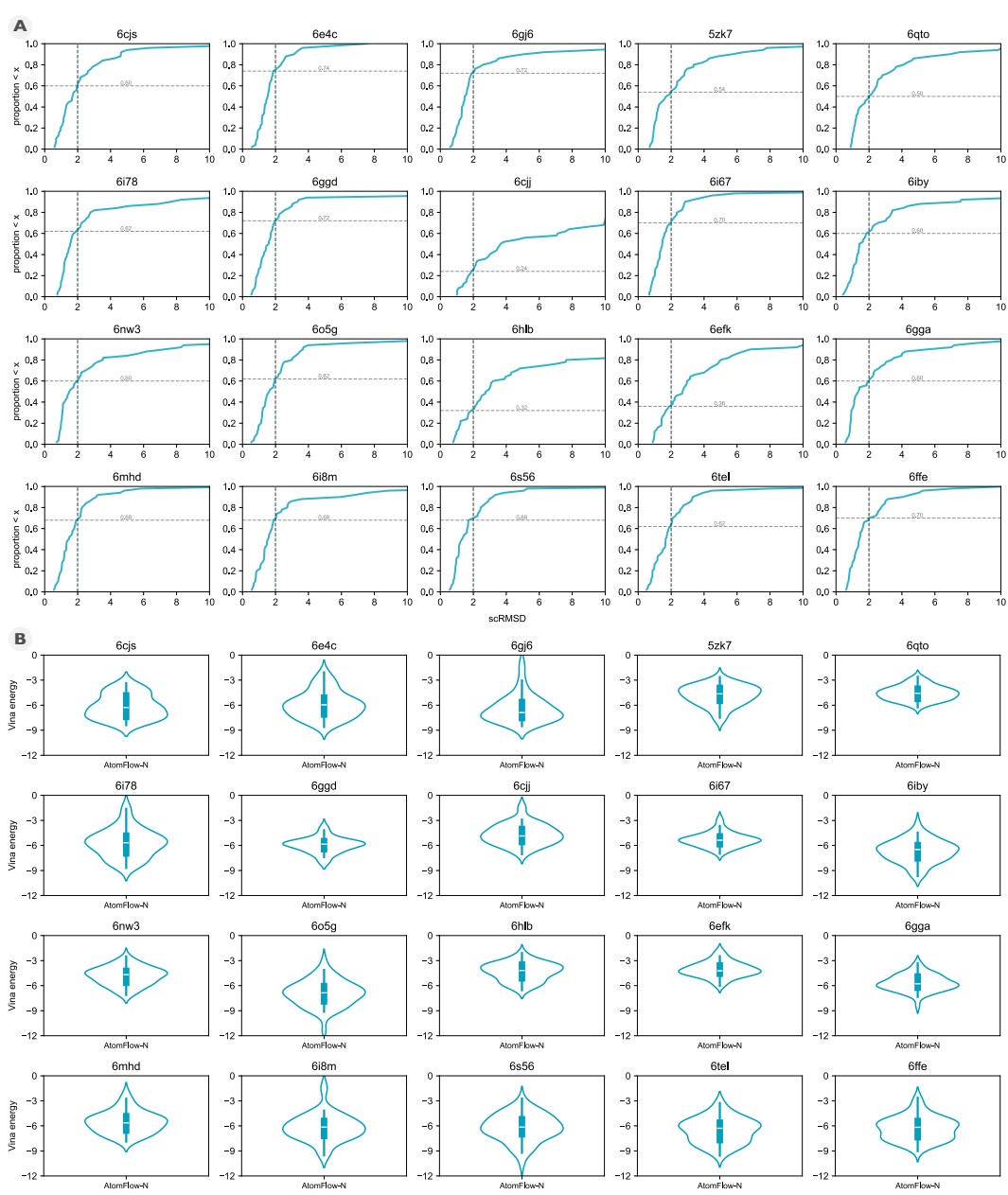

Figure 8: A: scRMSD of designs for each ligand in the extended set; B: Vina energy of designs for each ligand in the extended set.

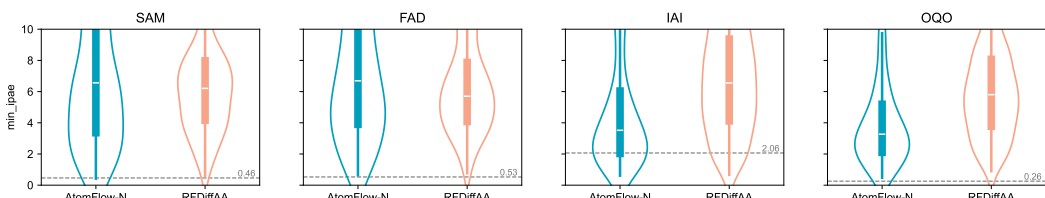

Figure 9: `min_ipAE` distribution of the generation results in the affinity experiment of the main text. The result of the native binder is displayed as a grey line.

| Ligand | AtomFlow (r=inf) | PG (r=3.5) | PG (r=5) | PG (r=6.5) | PG (r=8) | PG (r=9.5) |
|--------|------------------|------------|----------|------------|----------|------------|
| FAD | 0.79/3.74 | 7.10/7.38 | 6.75/7.81 | 7.29/8.35 | 20.92/24.23 | 23.12/25.23 |
| SAM | 0.83/2.01 | 2.12/2.62 | 2.77/2.99 | 2.94/4.03 | 12.39/14.49 | 13.79/14.74 |
| IAI | 0.56/1.82 | 0.71/0.85 | 0.95/1.02 | 2.04/2.28 | 3.59/5.53 | 9.02/11.71 |
| OQO | 0.59/1.63 | 1.20/1.26 | 1.70/1.79 | 2.40/2.45 | 11.59/11.94 | 2.13/2.41 |

Table 2: For this experiment, we used the natural binders of four ligands—FAD (7bkc), SAM (7c7m), IAI (5sdv), and OQO (7v11)—as input. To evaluate the design capability of PocketGen (PG) under different constraints, we progressively increased the design radius (minimum distance to ligand) from 3.5 to 9.5. The masked target area expanded with the radius, requiring the model to redesign increasingly larger regions of the protein. When the radius exceeded the protein's dimensions (radius greater than the protein size), all residues were masked, simulating our full-design setting. The table below presents the min/median scRMSD values for designs generated by PocketGen at each radius. For reference, scRMSD < 2 is generally considered a successful design. Notably, PocketGen's performance deteriorated significantly as the radius increased, reflecting its reliance on template residues. (At radius=8 for OQO, PocketGen generated designs with several residues misaligned with the ligand, leading to abnormally high scRMSD values.) Additionally, PocketGen does not support radius settings beyond 10, preventing direct simulation of ATOMFLOW 's fully template-free design scenario. The results of ATOMFLOW is derived from our main experiment.

| | QED | SA | LogP |
|--------|-----|-----|------|
| AtomFlow | 0.458±0.292 | 0.683±0.160 | 0.559±4.030 |
| PDBBind | 0.429±0.246 | 0.696±0.144 | -0.311±3.240 |

Table 3: QED, SA, and LogP of AtomFlow generated structures and PDBBind structures.

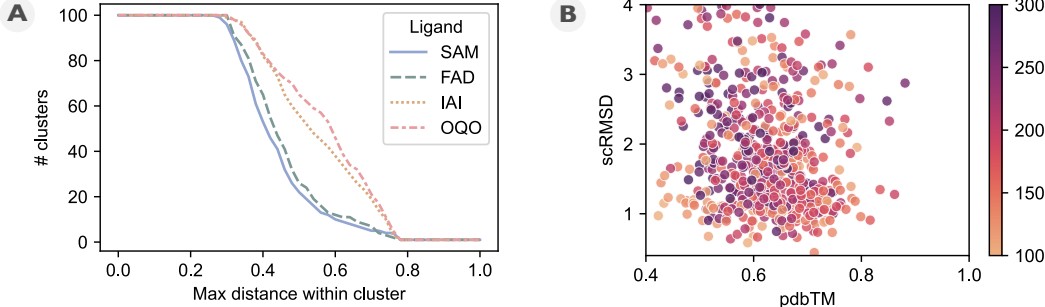

Figure 10: **A**: Cluster count based on different thresholds for the maximum difference within the cluster for each ligand in the evaluation set. **B**: Scatter plot of designability (scRMSD) vs. novelty (pdbTM) for ligands in the evaluation set.

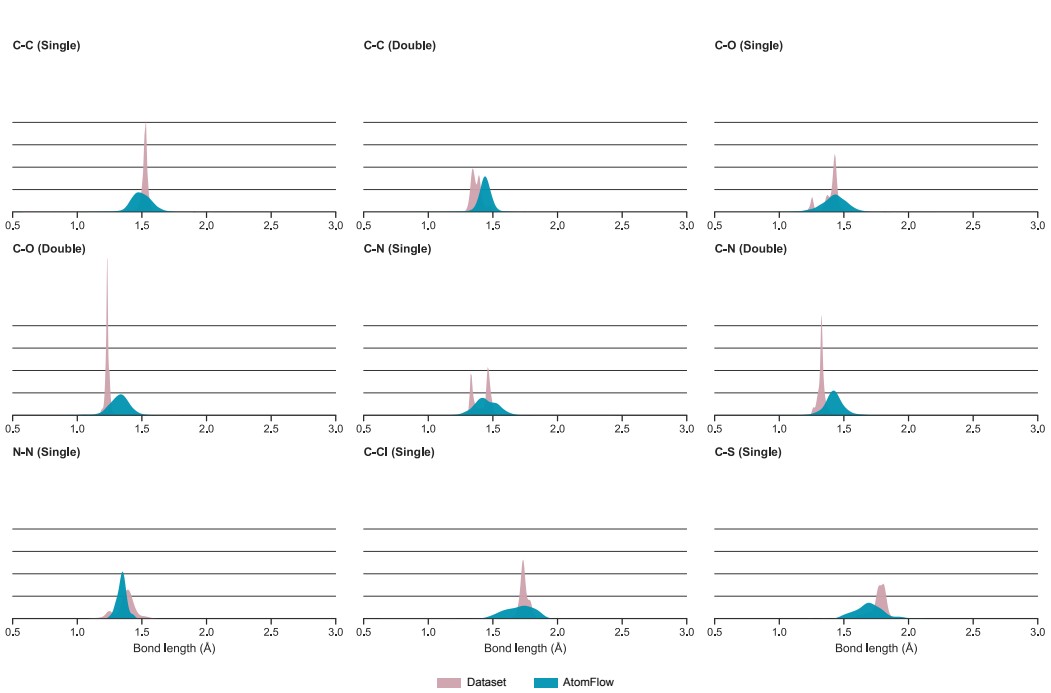

Figure 11: Chemical bond distribution of AtomFlow generated ligands for the extended set and ground truth ligands in the PDBBind dataset.

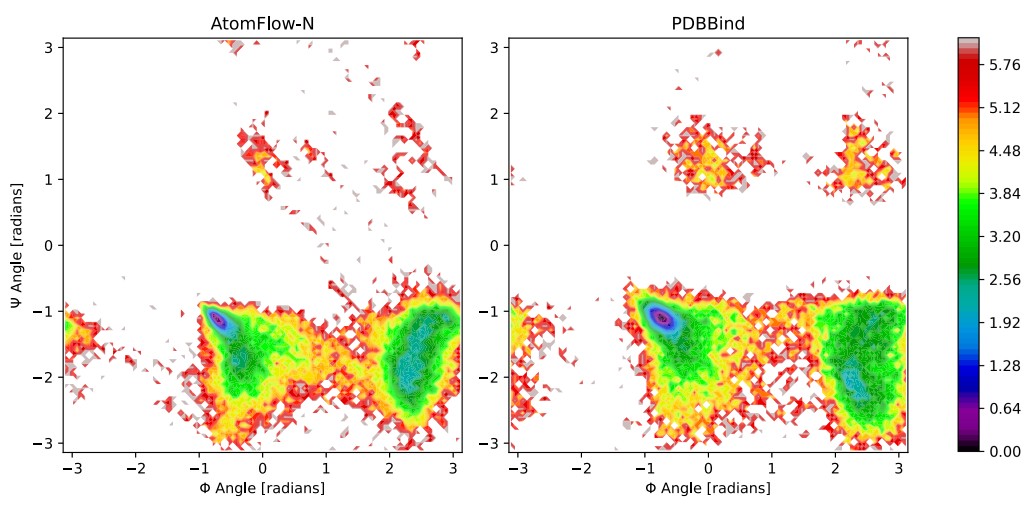

Figure 12: The Ramachandran plots for the generated protein (left) and the PDBBind protein (right), which demonstrate comparable **coverage** in the primary secondary structure regions.