# OpenReview forum: "Design of Ligand-Binding Proteins with Atomic Flow Matching"
_ICLR.cc/2025/Conference — Submitted to ICLR 2025_

### Official Review · Reviewer_iafd · 2024-11-02

**Soundness:** 3
**Presentation:** 4
**Contribution:** 3
**Rating:** 6
**Confidence:** 3

**Summary:**

In this work, the authors present ATOMFLOW, a deep generative model based on a flow-matching framework for designing ligand-binding proteins from 2D target molecular graphs. ATOMFLOW operates on representative atoms of biological tokens to capture ligand flexibility, iteratively generating ligand conformations and protein backbone structures. Overall, this paper introduces an innovative model that offers a novel approach to designing proteins that bind to specific molecules.

**Strengths:**

-  The paper is well-structured and logically organized. Readers can easily follow the authors’ thought process, and the content is presented in a way that is easy to understand.
- Conceptualizing both proteins and molecules as biotokens with representative atoms is an interesting approach. Although similar concepts are mentioned in Umol and RoseTTAFold All-Atom, using this framework to design specific proteins without relying on the initial ligand structure demonstrates impressive performance and is commendable.
- Compared to RFDiffusionAA, ATOMFLOW shows competitive results in specific protein-molecule design tasks as well as in binding affinity prediction. The supplementary experiments in Table 8 of the appendix provide additional support for the current experimental results.

**Weaknesses:**

- The intervals used in modeling the distance map for ligands and proteins are not uniformly divided. While the authors account for differing precision requirements between residues and atoms, given that the paper frames proteins and molecules as biotokens with representative atoms, could the distance partitioning criteria be standardized? Establishing a more universal distance characterization approach—without introducing additional biases—might provide a more consistent method for distance representation.
-  The authors mention that “the model is first trained on solely generating the protein structure for 40k steps.” In this phase, is ligand information completely omitted, or is it set as specific input information? Further details on the full training process would be beneficial.
- The paper indicates that, during training, the FAPE loss is divided into protein-protein interaction, protein-ligand interaction, and ligand-ligand interaction components. However, these three losses are not directly introduced or explained in detail. Could the authors elaborate on the definitions of these three loss components and how they are calculated?
- The paper appears to lack ablation studies that could support certain design choices in the model. For instance, what impact would removing the first phase of training (the 40k steps on protein structure generation only) have on model performance? Additionally, how were the initial training and fine-tuning parameters in the second training phase determined, and what criteria guided the transition steps?

**Questions:**

- As I mentioned in the weakness, I hope the author can add some ablation experiments to demonstrate the effectiveness of some training strategies used in the paper.
- The author can provide a more detailed explanation of the composition of the loss in Section 4.4. I haven't seen relevant information in the appendix yet.

---

> ### Author Response · Authors · 2024-11-19
> **A kind rebuttal to reviewer iafd**
>
> We really appreciate your valueable feedbacks on our paper. Thank you for affirming the technical contribution and the overall experiment results. We address the issues you discussed below, and hope the explanation may lead to a more positive judgement of our paper.
>
> ### Q1. A unified distance encoding.
> Thanks for pointing out a possible enhancement of the distance encoding in AtomFlow. During our training, we find that the model is quite capable of dealing with the uneven split of binning. On the contrary, directly provide the original distance resulted in unsatisfactory training trajectory. We hypothesize that a more universal distance encoding should encode embed the raw distance into a high-demensional vector, and this is a great direction for future exploration.
>
> ### Q2. Training details.
> We've added more training details to the appendix. Regarding your question, ligand tokens are not given during the first training stage. The first stage trains an unconditional protein generation model, while the second stage turns it to a conditional protein binder + ligand conformer generation model.
>
> ### Q3. On the training objective.
> We've added the following explanation to the appendix. The FAPE loss is defined as an average of all pairs of tokens in the original paper, so the calculation process first yield a FAPE matrix and then produce the average value of the matrix. The protein-protein, protein-ligand and ligand-ligand loss calculates the average value of the sub-matrixs defined as (row: protein, col: protein), (row: protein, col: ligand), and (row: ligand, col:ligand).
>
> ### Q4. On the ablation.
>
> First, we sincerely apologize for a typo in the original text. The training steps for the two stages should be 400k and 300k, rather than 40k and 30k as originally stated. This error has now been corrected.
>
> Regarding the ablation, since training a protein design model is significantly time-consuming, the design choices of our training strategy is largely determined by grid searching possible design space and save the training trajectory of the first 30~50k steps. We compare the training trajectories and select the best configuration that meets the following criteria:
>
> - a) The final distogram loss should close the minimum we get among the configurations (around 2.0).
>
> - b) The L_CFM-FAPE should not decline too fast at the first 10k steps. The first 10k steps is for the transformer stack to learn a relatively steady output, indicated by the decline of the distogram loss. A decline of L_CFM-FAPE at this stage will resulted in an undesired local minimum. Then L_CFM-FAPE should decline fast right after the distogram loss turns to decline much smoother. We select the configuration with the lowest L_CFM-FAPE at the end of training.
>
> We decide the end of each training stage when the training converges, with the following criteria:
>
> - a) the decline rate of every single loss is small.
>
> - b) the structural violence of sampled structures (counts of CA atom violation) converges.
>
> An initial study on directly train the second stage shows unsatisfactory training trajectory. Since the ligand conformer is way easier to generate compared to protein folds, the FAPE loss declines too fast even before the distogram loss, resulted in unstable TransformerStack output, and leading to a diverge of the model after ~30k steps. The resulted model with minimum loss is able to predict the ligand structure, with random protein residue position, which is unusable.
>
> Due to resource limit, we could not provide a comprehensive ablation study since only the configuration we selected has been trained to the end. We hope the details regarding our training process provide enough information to justify our decisions.
>
> We also want to share a new experimental result with you. During the review process, several AlphaFold 3 replications, including the original models, were released. Leveraging these AlphaFold 3-like model predictions, we developed what is likely a more reasonable in silico metric for evaluating binding affinity. On this metric, AtomFlow outperforms RFDiffusionAA. The detailed results have been included in the appendix of our paper, and a more comprehensive analysis will be provided in the final version. We hope this introduces an additional evaluation approach that can benefit the community.
>
> We hope our explanation has addressed your questions and concerns thoroughly. We sincerely appreciate your valuable feedback and are always happy to clarify further or discuss ways to improve our work. If you find our responses satisfactory, we would be grateful for your consideration of a higher score. Thank you!

---

> > ### Comment · Reviewer_iafd · 2024-11-24
> > **Thanks for the rebuttal**
> >
> > Thanks for the comprehensive response and I will maintain my positive score.

---

### Official Review · Reviewer_NTy9 · 2024-11-02

**Soundness:** 3
**Presentation:** 3
**Contribution:** 2
**Rating:** 5
**Confidence:** 4

**Summary:**

This paper presents AtomFlow, a flow matching model for designing a protein structure to bind a small molecule ligand. The model jointly denoises the structure of the protein and ligand and thus does not require knowledge of the ligand pose, unlike RFDiffusionAA. The architecture is based on AlphaFold and predicts denoised structures from a distance map input of the input structure. The method is evaluated on (1) the set of 4 ligands studied in RFDiff-AA, and (2) an expanded set of ligands curated by the authors. AtomFlow is shown to have comparable or better designability than RFAA as well as similar Vina score.

**Strengths:**

* The work is solidly executed, with sensible architectural choices, strong initial evaluations, and clear and concise writing.
* The task tackled is significant and the competitive results signify well-executed model engineering and training practices.
* The authors re-derive the quotient-space flow matching fromework from AlphaFlow with more solid theoretical justification.
* The paper is quite clearly written. The figures are well made, visually appealing, and informative.

**Weaknesses:**

**Originality**
* The methodology can be described as a flow-matching version of RFDiff-AA and does not score high on originality / novelty from a ML perspective. Further, the flow model architecture and noising process are based on AlphaFlow, with different justification but no difference in practice as far as I can tell. To improve on this axis, while it's not clear that more methodological novelty is needed for its own sake, the authors could focus on novel evaluations or applications of the proposed method.

**Quality**
* The computational evaluations are well executed, but limited in scope. Most of the analysis focuses on only 4 ligands, raising concerns about sample size and statistical significance.
* The diversity and novelty evaluations are nice, but only AtomFlow is evaluated, not RFDiff-AA or the other baselines.


**Significance**
* The overall significance of the contribution is unclear as it represents an incremental methodological advance over RFDiff-AA with more or less the same model capabilities. The authors argue that not needing to specify the ligand pose is a big plus, but no meaningful evidence or use case is provided for this distinction. After all, RFDiff-AA has been experimentally validated, whereas AtomFlow-generated poses have not. It is of course not expected for a ML submission to experimentally validate the proteins, but it should be made a bit clearer why the main point of difference with RFDiff-AA is important to tackle as a ML problem.

**Questions:**

No specific questions.

---

> ### Author Response · Authors · 2024-11-19
> **A kind rebuttal to reviewer NTy9 (Part 1/2)**
>
> Thank you for your detailed feedback on our submission. We appreciate your insights, particularly your concerns regarding novelty and the sufficiency of experiments, which may have influenced your current assessment of our work. In this rebuttal, we aim to address these points comprehensively and provide further evidence to clarify the contribution of our work.
>
>
> ### 1. The contribution of our work
>
> **a. Significance of the Problem**
>
> AtomFlow seeks to bypass the limitation of requiring a fixed binding pose of ligand in RFDiff and RFDiffAA, since the bound structure of ligand is often unknown in our designing practice. A bound conformer usually diverges greatly from the ideal structure, which is shown in the introduction of our paper, and it's difficult to determine which pose will be optimal before design since it's determined by the binder [1]. In real practice, a known bound conformer is usually unavailable for a novel target [2], which greatly limits the adoption of RFDiffAA in the workflow.
>
> **b. Novelty of our Methodology**
>
> We proposed a novel general framework for an important biological problem, which is built on top of AlphaFold 2 but with non-trivial augmentation and adaptation. AtomFlow address the problem of co-modelling protein and small molecule ligands, by introducing a more generalized biotoken representation and a probability flow model on the representative atoms. We acknowledge that the final form of our probability flow is similar to AlphaFlow. However, the derivation, noise form, and final loss is different from theirs. Furthermore, **AlphaFlow is designed to generate multiple protein ensembles for a FIXED sequence**, which means that its generation process is largely restricted by the given protein sequence, and the flow matching process is for increasing the diversity of AlphaFold prediction, rather than modelling the distribution of all possible folds. We've tested the original AlphaFlow code on general protein design and find that it could not properly generate a fold without specifying a proper sequence.
>
> **Though not highlighted in the paper, we're actually the first to prove that it's possible to train a model to generate arbitrary protein fold with an SE(3)-invariant loss function**, rather an MSE loss used in FrameDiff, Genie and AlphaFold 3 [3,4,5,6]. An SE(3)-invariant loss gives the same supervising signal for a predicted structure under arbitrary SE(3) transformation, while an MSE loss requires the model to generate a structure with a specific global translation and rotation, falsely punishing similar structure under different SE(3) transformation, which we believe is suboptimal. AlphaFlow partially proves this, but some issues related to their derivation leads to unsatisfactory results when training a design model without sequence. Our derivation has a much better theorectical ground and leads to satisfactory model performance, proving this is a working route.
>
> Furthermore, **we enhanced our evaluation according to your suggestion by adding an alternative in silico metric for binding affinity based on minimum interchain pAE predictions to the appendix**, inspired by AlphaProteo [7].  During the review process, several AlphaFold 3 replications, including the original models, were released. Though we're not abel to run the AlphaFold 3 model locally due to a licensing issue, we developed an alternative in silico metric based on Chai-1 [8], calculating the minimum value across all interchain terms in the PAE matrix (min_ipAE, lower is better). Note that this metric is proved to be a good indicator for protein-protein binder design, but not validated on small molecule-protein binders. We've added the results of AtomFlow-N and RFDiffAA to the appendix. The results shows that the AtomFlow-generated binders have lower min_ipAE than the ones from RFDiffAA, and both models have the ability to generate binders with similar min_ipAE as natural ones. We're working to develop better metrics as in silico proxy for ligand-binding protein design based on wet lab verification.

---

> > ### Author Response · Authors · 2024-11-19
> > **A kind rebuttal to reviewer NTy9 (Part 2/2)**
> >
> > ### 2. The Experiment
> >
> > **a. Sufficiency of Experiment**
> >
> > We mainly focus on four ligands in the main text to align with RFDiffAA. However, we agree with you that only four ligands may not be enough to prove its performance, and the original paper provides the results for an additional evaluation set in the appendix. The addtional evaluation is conducted on 20 ligands and shows similar patterns as discussed in the main text. We also refer you to our reply to reviewer Z34b (2.a), where we provide the generation results for the L1000 set in CASP16.
> >
> > **b. Lack of Comparison**
> >
> > Thanks for pointing out the lack of comparison in the diversity and novelty experiment. The form we report the diversity and novelty follows existing literature [3]. We've added the results of RFDiffAA to the appendix, though these are not metrics that we mainly focus on optimizing, since our goal is to generate a valid binder for arbitrary ligand with acceptable diversity (not always the same) and novelty (not simply copying structures from PDB). The diversity of AtomFlow designs is better than RFDiffAA, while the AtomFlow generated results tends to be more conservative in terms of pdbTM novelty. We believe this is because we didn't train AtomFlow on a full training set including all PDB structures and the distillation data. This is our future work and we'll release an updated model once available.
> >
> > We hope our explanation has addressed your questions and concerns thoroughly. We sincerely appreciate your valuable feedback and are always happy to clarify further or discuss ways to improve our work. If you find our responses satisfactory, we would be grateful for your consideration of a higher score. Thank you!
> >
> > [1] Corso, G., Stärk, H., Jing, B., Barzilay, R., & Jaakkola, T. (2022). Diffdock: Diffusion steps, twists, and turns for molecular docking. arXiv preprint arXiv:2210.01776.
> >
> > [2] Bick, M. J., Greisen, P. J., Morey, K. J., Antunes, M. S., La, D., Sankaran, B., ... & Baker, D. (2017). Computational design of environmental sensors for the potent opioid fentanyl. Elife, 6, e28909.
> >
> > [3] Yim, J., Trippe, B. L., De Bortoli, V., Mathieu, E., Doucet, A., Barzilay, R., & Jaakkola, T. (2023). SE (3) diffusion model with application to protein backbone generation. arXiv preprint arXiv:2302.02277.
> >
> > [4] Yim, J., Campbell, A., Foong, A. Y., Gastegger, M., Jiménez-Luna, J., Lewis, S., ... & Noé, F. (2023). Fast protein backbone generation with SE (3) flow matching. arXiv preprint arXiv:2310.05297.
> >
> > [5] Lin, Y., Lee, M., Zhang, Z., & AlQuraishi, M. (2024). Out of Many, One: Designing and Scaffolding Proteins at the Scale of the Structural Universe with Genie 2. arXiv preprint arXiv:2405.15489.
> >
> > [6] Abramson, J., Adler, J., Dunger, J., Evans, R., Green, T., Pritzel, A., ... & Jumper, J. M. (2024). Accurate structure prediction of biomolecular interactions with AlphaFold 3. Nature, 1-3.
> >
> > [7] Zambaldi, V., La, D., Chu, A. E., Patani, H., Danson, A. E., Kwan, T. O., ... & Wang, J. (2024). De novo design of high-affinity protein binders with AlphaProteo. arXiv preprint arXiv:2409.08022.
> >
> > [8] Chai Discovery, Boitreaud, J., Dent, J., McPartlon, M., Meier, J., Reis, V., ... & Wu, K. (2024). Chai-1: Decoding the molecular interactions of life. bioRxiv, 2024-10.

---

> > > ### Author Response · Authors · 2024-11-25
> > > **Follow-Up on Rebuttal Response**
> > >
> > > Dear Reviewer NTy9,
> > >
> > > Thank you for taking the time to review our paper and for your valuable feedback.
> > >
> > > We hope we have addressed your concerns in our rebuttal and would greatly appreciate it if you could let us know whether there are any remaining questions or issues to address, as the review deadline is approaching. If the rebuttal resolves the issues you mentioned, we kindly ask you to consider updating your evaluation to reflect this.
> > >
> > > Thank you again for your effort and time. We look forward to your response!

---

> ### Comment · Reviewer_NTy9 · 2024-11-26
>
> Thanks for the detailed response and updated experiments. My primary concern is that, with only 4 ligands and a minor tweak in problem formulation from RFDiff-AA, the paper does not sufficiently demonstrate a value-add on top of RFDiff-AA. It should be noted that 4 experimental validations are very different from 4 in silico successes, as experimental success is what actually matters and in silico metrics are just a noisy approximation for that. This is not to say I am not sympathetic to the authors' claims; quite to the contrary, I believe they may very well have a model on their hands that is better than RFDiff-AA. However, with the current limited evaluations, statistical significance is hard to judge, and publication may be premature.

---

> ### Author Response · Authors · 2024-11-26
> **Regarding your concerns**
>
> Thank you for your thoughtful feedback! We truly appreciate your recognition of our model's contributions and are dedicated to thoroughly addressing your concerns in the following sections.
>
> ### 1. Evaluation on Additional Dataset
>
> We emphasize that the initial draft of our paper **already included an evaluation** on an **additional dataset of 20 ligands** in the appendix, with atom counts ranging from 21 to 104. During the rebuttal stage, we expanded our evaluation metrics to include **iPAE for binding affinity, geometric bond distribution, Ramachandran plot, and chemical validity metrics including SA, QED, and LogP**. These additional evaluations further strengthened our results. The extended results corroborate the trends in the main text (as highlighted by Reviewer IAfd) and further validate the robustness of AtomFlow’s outputs.
>
> > **Note:** Extending experiments is a resource-intensive process, as a comprehensive generation for different lengths is required for each ligand, making the process significantly more demanding than evaluating unconditional design models. We are working towards incorporating even more comprehensive supplementary evaluations in the final version.
>
> We acknowledge that our paper and previous rebuttal may not have sufficiently emphasized the additional evaluation in the appendix, and we hope that our response addresses this gap clearly.
>
> ### 2. Future Work on Laboratory Validation
>
> We agree that laboratory success is the ultimate benchmark for a design model. While this is beyond the scope of the current study, it is a key focus for our future work.
>
> ### 3. On RFDiffusionAA & Our Contribution to the ML Community
>
> We recognize the exceptional utility of RFDiffusionAA in protein design, and the primary goal of this work is not to replace RFDiffusionAA.
>
> From **a machine learning perspective**, AtomFlow introduces a **user-friendly tool with a solid theoretical foundation** in the SE(3)-invariant quotient space of conformer structures for designing binders to entirely new targets with unknown ligand conformers (as supported by Reviewer Z34b). AtomFlow incorporates a **unified, novel biotoken-based atomic flow matching framework** for designing flexible interactions. Additionally, we are the first to **demonstrate the effectiveness of SE(3)-invariant flow** in de novo design.
>
> Inspired by recent advances in protein-protein binder design (e.g., AlphaProteo), we **introduced the iPAE prediction of AlphaFold3-like models** as part of our binding affinity evaluation. During the rebuttal stage, we **significantly improved our supplementary experiments**, providing insights for future machine learning researchers and further enhancing our contributions. These aspects mark AtomFlow as a complementary and valuable tool for the community.
>
> We hope this response clarifies our contributions and addresses your concerns comprehensively. Thank you once again for your insights.

---

> > ### Comment · Reviewer_NTy9 · 2024-11-26
> >
> > I see the 20 additional ligands in the appendix; however, since there are no comparisons with RFDiffusion-AA on those ligands, these experiments do not support the claim that AtomFlow (and its problem formulation) provides additional value-add over existing methods.

---

> > > ### Author Response · Authors · 2024-11-28
> > > **Additional Results and Clarification**
> > >
> > > Thank you for pointing out the lack of baseline comparisons in our supplementary experiment.
> > >
> > > The bottleneck for a comprehensive evaluation in this task lies in computational resources. We noticed that RFDiffusionAA, as stated in their paper, also performed in silico testing on only four ligands. As we previously mentioned, conducting a complete evaluation over a broader range of ligands demands significant computational resources. To clarify this point, we provide a runtime experiment for your reference:
> > > > We conducted experiments to generate samples for the ligand FAD using AtomFlow and RFAA, with amino acid lengths of 100, 150, 200, 250, and 300. For each length, we measured the time (**in minutes**) required to generate a single sample. Each experiment was repeated three times, and we reported the average time along with the standard error. During the experiments, each method had exclusive access to its respective GPU.
> > >
> > > | Method     | L=100             | L=150             | L=200             | L=250             | L=300             |
> > > |------------|-------------------|-------------------|-------------------|-------------------|-------------------|
> > > | AtomFlow   | 0.49 ± 0.0       | 0.51 ± 0.01       | 0.58 ± 0.01       | 0.79 ± 0.01       | 0.88 ± 0.0       |
> > > | RFAA       | 2.48 ± 0.03       | 2.75 ± 0.04       | 3.23 ± 0.01       | 4.02 ± 0.04       | 4.58 ± 0.05       |
> > > | Speedup    | 5.06x                | 5.39x                | 5.57x                | 5.09x                | 5.2x                |
> > >
> > > On average, RFDiffusionAA requires 5 times the time taken by AtomFlow to generate a sample. While result quality is far more important than runtime efficiency in the field of protein design, **we believe that this speed advantage of AtomFlow provides additional value-add over RFDiffusionAA**, which significantly accelerates the evaluation and generation. However, this disparity in runtime makes a full baseline evaluation very challenging.
> > >
> > > This is why we chose to only report our own performance on the extended ligand set similar to the fully evaluated and compared 4 ligands to demonstrate that it performs comparably to RFDiffusionAA while handling scenarios without requiring input ligand conformers. That said, we are actively working to secure the necessary computational resources to complete a full evaluation, and **a full supplementary evaluation will be included in the final version**.
> > >
> > > To address your concern in the interim, **we have conducted an extended evaluation of RFDiffusionAA** for amino acid length = 200. The comparative **results are available** at https://pasteboard.co/giZfk1LTrpfz.jpg , which confirm the trends observed in the four-ligand evaluation and further validate AtomFlow's performance.
> > >
> > > We hope this response addresses your concern adequately.

---

### Official Review · Reviewer_Z34b · 2024-11-02

**Soundness:** 2
**Presentation:** 3
**Contribution:** 2
**Rating:** 6
**Confidence:** 5

**Summary:**

The authors present the ATOMFLOW architecture for the design of protein folds conditioned by the ligand SMILES description. The tool actually can be useful for the bioinformatics community, but many more validation tests are required here. I also thank the authors for providing the code, we had fun time generating different folds and assessing them.

**Strengths:**

It's a well written technical paper about a novel architecture that can be useful for the bioinformatics community. I am thankful to authors for providing the code. Please remove the infinite loop and the check for the clash score, it takes too much time.

We have executed it on a couple of example, including recent L targets from CASP. I have carefully examined the generated output for L1000 ligand set. With a significant probability (about 30%), AtomFlow finds the native experimental fold (we have ~20 PDB structures with ligands different from those in CASP) and the binding site, and is also rather stable with variations of the sequence length. If I vary the design length slightly, AtomFlow seems to prefer to shorten loops even those inside the binding site, so it seems it has memorized the space of all protein folds in the PDB and simply selects the one it has seen during training. It actually can be a useful tool for fold search and classification.

Individual protein-ligand contacts vary in quality significantly. So I have doubts in any usefulness of further affinity prediction providing a very low quality of experimental affinity data (see below).

We have also tried to generate folds for a purine ligand, which is a part of nucleotide bases (expecting many different folds to be generated). Surprisingly, at a constant generation length, the results were extremely conservative, with zero novelty and extremely low diversity, preferring the fold that dominantly occurs in the PDB. So, I suggest the authors to run more cross-validation tests hiding some fold classes from the training and trying to reproduce them in the test. Otherwise it seems there is a significant bias towards the folds that are abandon in the PDB.

**Weaknesses:**

There is little technical novelty in the architecture blocks, loss function of flow matching process (but the architecture is novel and useful).

Since you are using the FAPE loss, why do you need equations 3-5, references to Riemannian Flow Matching, and fancy words? It will be much cleaner to remove this part.

Binding affinity experiments are far from complete and better to be removed. You need to split the data correctly and run multiple affinity benchmarks, for example, following CASF and DUDe protocols. It will also be useful to provide energies and folds for some well-studied proteins that have multiple structures with different ligands (they are currently coming in the PDB, you may look at the last CASP experiment). Please also provide Vina scores for native ligand poses. Vina is not the best proxy for the affinity, as it was design only to rank binding poses for the same protein-ligand complex.

Please also be aware of the quality of current affinity training sets -- e.g., https://pubs.acs.org/doi/full/10.1021/acs.jcim.4c00049.

"The scatter plot of scRMSD vs. pdbTM shown in Figure 6B reveals that ATOMFLOW has the ability to generate structures that are quite different from existing proteins with acceptable designability" - Well, all of your designs have scores > 0.5, a few are around 0.5. It means that the fold structure is the same as in the PDB, no new folds have been discovered. Please try to discover a fold with a TM score around 0.2.

**Questions:**

LCFM and LCFM-FAPE indeed give zero at the same point. But why their gradients will be similar?

Can you explain more what does it mean "since the aligning object x varies upon training."? It contradicts all the theory you have just presented, no?

Can you comment on proper cross-validation experiments by removing certain fold classes from training? The same is valid for affinity experiments. Please also consider novel data with one protein and multiple ligands as validation, see four datasets from CASP16, for example.

---

> ### Author Response · Authors · 2024-11-19
> **A kind rebuttal to reviewer Z34b (Part 1/2)**
>
> Thanks for your review on our work.
> We are really happy to have a feedback from your hands-on experience of AtomFlow, and we are glad to see that AtomFlow successfully designs protein binders for your selected ligands. We address each of your concerns or the issues encountered below.
>
> ### 1. The infinite loop.
>
> We'll remove the infinite loop in the final release.
>
> ### 2. The diversity and novelty of designs.
>
> **a. Diversity.** We replicated your encountered situation that AtomFlow consistently generates one same fold for a given ligand in our released version, but the version used in evaluation seems not showing similar phenomenon. This should be a bug and will be fixed in the final release. We've generated 10*5 binders each for the L1000 set in CASP16 with lengths in [100, 150, 200, 250, 300] and uploaded them to https://anonymous.4open.science/r/AtomFlow-suppl-data-7384/L1000_results.zip .
>
> **b. Novelty.** We admit that the overall designs generated by AtomFlow tends to be conservative. After intensive search, the most novel fold found by AtomFlow in our experiment has pdbTM=0.347. We believe there's space for us to explore to increase the novelty of designs, and thank you for pointing out the novelty issue. We're trying to improve the design novelty of our model, perhaps by adding more training data and designing a better training strategy.
>
> ### 3. The affinity results.
>
> **a. Energy function as a proxy.** Thanks for pointing out that Vina is not a best proxy of affinity. It seems that you somewhat misunderstood the motivation of our affinity experiment. Both AtomFlow and RFDiffAA are not designed to predict the binding affinity, and the results we report is an in silico calculation on the structures they generate. The experiment is to confirm that AtomFlow is really generating binders for the given ligand, rather than generating a random fold without considering the target. We know that there does not exist an in silico metric that perfectly aligns with real binding affinity, but a metric is needed and we follow the practice of exising literature [1,2] to adopt an energy calculated by docking method as a proxy. We also thank you for suggesting adding the ground truth value of Vina energy to the paper, and we'll update the figures in our paper. The ground truth energy for the four ligands are: IAI (-9.864), FAD (-13.459), SAM (-7.115), OQO (-9.594). AtomFlow successfully designs binders with lower Vina energy than ground truth for IAI, SAM and OQO. No methods (including RFDiffAA) generate binders with lower Vina energy than ground truth for FAD.
>
> **b. Data splitting (Your Question 3).** We properly splitted the PDBBind dataset following the official split, following DiffDock [3]. FAD and SAM are in the training set, while IAI and OQO are not. We adopt this split since the main purpose of AtomFlow is to generate binders for ligands no matter seen or not. We believe splitting data in a way that hides several folds from training and see whether the model can generate an unseen fold is a very interesting research direction. However, this is beyond the scope of our research and should better start from addressing the novelty issue on unconditional protein design.
>
> **c. AlphaFold 3-like model as an affinity proxy.** Inspired by your comments on Vina energy, we feel it necessary to introduce an additional in silico proxy for binding affinity since the paper will set up a standard for future research to follow. We noticed that AlphaProteo [4] adopts several metrics produced by AlphaFold 3 as in silico filters. At the time we finished our draft, there's no publicly available AlphaFold 3 for us to run locally. Recently, several AlphaFold 3 replications and the original AlphaFold 3 are released. Though we're not abel to run the AlphaFold 3 model locally due to a licensing issue, we developed an alternative in silico metric based on Chai-1 [5], calculating the minimum value across all interchain terms in the PAE matrix (min_ipAE, lower is better). Note that this metric is proved to be a good indicator for protein-protein binder design, but not validated on small molecule-protein binders. We've added the results of AtomFlow-N and RFDiffAA to the appendix. The results shows that the AtomFlow-generated binders have lower min_ipAE than the ones from RFDiffAA, and both models have the ability to generate binders with similar min_ipAE as natural ones. We're working to develop better metrics as in silico proxy for ligand-binding protein design based on wet lab verification.

---

> > ### Author Response · Authors · 2024-11-19
> > **A kind rebuttal to reviewer Z34b (Part 2/2)**
> >
> > ### 4. On referencing the RFM.
> > AtomFlow is not trained with the original FAPE loss since it adds an additional weight for different timestamp, which differs from existing works that trains a flow matching model directly on FAPE loss [6]. The form of the final loss is derived from RFM, with solid theoretical ground.
> >
> > **(Your Question 1&2)** The gradient of L_CFM and L_CFM-FAPE are not similar as they do lead to different training trajectory. The aligning target in L_CFM is the noisy input structure, which is different at each training step, even for the same structure. Our initial experiment shows that L_CFM leads to suboptimal training trajectory, so we replaced it with L_CFM-FAPE. By splitting the L_CFM into time-related weighting and time-dependent model performance, we are able to show that L_CFM-FAPE is an approximation of L_CFM. Generally, predicted structures get higher loss when they diverge from the ground truth, but two structures with similar divergence may have different ordering in terms of how diverge they are based on L_CFM and L_CFM-FAPE.
> >
> > We hope our explanation has addressed your questions and concerns thoroughly. We sincerely appreciate your valuable feedback and are always happy to clarify further or discuss ways to improve our work. If you find our responses satisfactory, we would be grateful for your consideration of a higher score. Thank you!
> >
> > [1] Zhang, Z., Shen, W. X., Liu, Q., & Zitnik, M. (2024). Efficient Generation of Protein Pockets with PocketGen. bioRxiv.
> >
> > [2] Krishna, R., Wang, J., Ahern, W., Sturmfels, P., Venkatesh, P., Kalvet, I., ... & Baker, D. (2024). Generalized biomolecular modeling and design with RoseTTAFold All-Atom. Science, 384(6693), eadl2528.
> >
> > [3] Corso, G., Stärk, H., Jing, B., Barzilay, R., & Jaakkola, T. (2022). Diffdock: Diffusion steps, twists, and turns for molecular docking. arXiv preprint arXiv:2210.01776.
> >
> > [4] Zambaldi, V., La, D., Chu, A. E., Patani, H., Danson, A. E., Kwan, T. O., ... & Wang, J. (2024). De novo design of high-affinity protein binders with AlphaProteo. arXiv preprint arXiv:2409.08022.
> >
> > [5] Chai Discovery, Boitreaud, J., Dent, J., McPartlon, M., Meier, J., Reis, V., ... & Wu, K. (2024). Chai-1: Decoding the molecular interactions of life. bioRxiv, 2024-10.
> >
> > [6] Jing, B., Berger, B., & Jaakkola, T. (2024). AlphaFold meets flow matching for generating protein ensembles. arXiv preprint arXiv:2402.04845.

---

> > ### Comment · Reviewer_Z34b · 2024-11-24
> >
> > I thank the authors for their comments.
> >
> > Personally, I've found the architecture useful for the biological community, but for them more tests need to be conducted regarding the foldability and novelty of the proposed folds.
> >
> > I thank the authors for the PAE experiments - this is probably the best in-silico available metric.
> >
> > I am not in a great favour of demonstrating Vina (or other) scores, especially those that look better than on crystal structures. The best thing you can probably do is to:
> > -- compare geometrical distributions to the ones in the training set (please demonstrate how well you generate from the trained sampes)
> > -- assess the chemical validity (provide crystallographic statistics) of the final protein-ligand complex.

---

> > > ### Author Response · Authors · 2024-11-25
> > > **Additional Metrics Added**
> > >
> > > Thank you very much for your valuable feedback. We truly appreciate your comments regarding the PAE metric, and we are grateful that you pointed out additional metrics that could be of interest to the biological community.
> > >
> > > In response, we have added the following results to the appendix, incorporating your suggestions:
> > >
> > > > All results presented below are based on our generated structures for the extended evaluation set in the appendix. This set includes a diverse collection of unseen ligands selected from the official test split of PDBBind.
> > >
> > > ### A. Geometrical Distributions
> > >
> > > We evaluated the common chemical bond lengths generated by AtomFlow in comparison to the ground truth bond lengths found in our training set. The results indicate that the ligands generated by AtomFlow exhibit geometric distributions highly similar to those of the ground truth data.
> > >
> > > > Result link (also available in the paper): [https://pasteboard.co/5yZGXjlAYkYV.jpg](https://pasteboard.co/5yZGXjlAYkYV.jpg)
> > >
> > > Furthermore, we assessed the generated structures by utilizing Ramachandran plots, which are fundamental tools for analyzing protein structures. Below are the Ramachandran plots for the generated proteins (left) and the PDBBind proteins (right). Both plots exhibit comparable coverage in the primary regions of the secondary structure, suggesting that the proteins generated by AtomFlow effectively capture the key structural characteristics of natural proteins.
> > >
> > > > Result link (also available in the paper): [https://pasteboard.co/VoNEvDVQKc8b.jpg](https://pasteboard.co/VoNEvDVQKc8b.jpg)
> > >
> > > ### B. Chemical Validity
> > >
> > > We also evaluated the generated ligands based on several key chemical validity metrics:
> > >
> > > -   QED: The quantitative estimation of drug-likeness, with values ranging from 0 (non-drug-like) to 1 (drug-like).
> > > -   SA: Synthetic accessibility, representing the difficulty of chemical synthesis, ranging from 0 (easily synthesized) to 10 (very difficult to synthesize).
> > > -   LogP: An important parameter characterizing the overall hydrophobicity of organic compounds.
> > >
> > > The table below summarizes the comparison between AtomFlow-generated ligands and those from PDBBind:
> > >
> > >
> > > | Metric             | QED           | SA            | LogP           |
> > > |--------------------|---------------|---------------|----------------|
> > > | AtomFlow-generated | 0.458 ± 0.292 | 0.683 ± 0.160 | 0.559 ± 4.030  |
> > > | PDBBind            | 0.429 ± 0.246 | 0.696 ± 0.144 | -0.311 ± 3.240 |
> > >
> > > These metrics indicate that AtomFlow-generated ligands maintain similar drug-likeness and synthetic accessibility as the PDBBind ligands, while demonstrating slightly different hydrophobic characteristics. This suggests that our method can effectively generate ligands with properties comparable to the ground truth dataset.
> > >
> > > Thank you once again for your insights. We hope these additional results address your concerns comprehensively, and we remain open to further suggestions. If our response has satisfactorily addressed your questions, we would be grateful if you could consider raising the evaluation score.

---

> > > > ### Comment · Reviewer_Z34b · 2024-11-25
> > > > **On additional metrics**
> > > >
> > > > I thank the authors but I am a bit confused here. The ligands (the SMILES strings) are given to the networks as input, you don't generate them, right? Then, please explicitly specify for the new metrics (QED, SA, LogP) if they take into account any 3D geometry.  If not, they are not very useful here.
> > > >
> > > > Ramachandran plots is a good quality indicator. You may also think of some clash statistics between ligands and the generated protein.

---

> > > > > ### Author Response · Authors · 2024-11-29
> > > > > **Clarification and New Results**
> > > > >
> > > > > We deeply appreciate your thoughtful feedback and thank you for your patience while waiting for our response.
> > > > >
> > > > > We carefully reviewed the previously provided *Chemical Validity* data and confirmed that the calculation of these metrics does not consider 3D structures. We acknowledge that using these metrics is not meaningful for evaluating the generated results in this context. Therefore, **we will remove these metrics and apologize for any confusion caused**.
> > > > >
> > > > > To further validate the chemical plausibility of the generated structures, we have selected PoseCheck [1], a toolkit developed as part of a benchmark for structure-based drug design. While our primary objective is to develop a ligand-binding protein, instead of drug design, we find their metrics valuable for assessing the interaction and chemical validity of the protein-ligand complex. We report the following three metrics:
> > > > >
> > > > > - **Clash** (lower is better): Evaluates the plausibility of protein-ligand binding poses by measuring the number of atomic pairs within a distance smaller than their van der Waals radii.
> > > > >
> > > > > - **Strain** (lower is better): Assesses ligand conformational plausibility by calculating the difference in internal energy before and after ligand relaxation.
> > > > >
> > > > > - **Interactions** (higher is better): Quantifies the number of chemical interactions formed in protein-ligand complexes, focusing on four types: Hydrogen Bond Acceptors, Hydrogen Bond Donors, Van der Waals Contacts, and Hydrophobic Interactions.
> > > > >
> > > > > The results are shown in https://pasteboard.co/8P1EHQYQCR5H.jpg , and they will be added to the final version of our paper.
> > > > >
> > > > > > The data is based on the same samples generated in the binding affinity experiment described in the main text, ensuring consistency with the experimental setup. For each sample’s eight different sequences, we selected the one with the least clash for statistics. Ground truth conformer data is represented by gray lines. Since RFDiffusionAA does not generate ligand conformers, strain values remain identical to the ground truth.
> > > > >
> > > > > The results indicate that both AtomFlow and RFDiffusionAA generate samples with metrics close to those of the ground truth. Notably, AtomFlow achieves comparable performance to RFDiffusionAA while not requiring any input of the bound conformer.
> > > > >
> > > > > [1] Harris, C., Didi, K., Jamasb, A. R., Joshi, C. K., Mathis, S. V., Lio, P., & Blundell, T. (2023). Benchmarking Generated Poses: How Rational is Structure-based Drug Design with Generative Models?. arXiv preprint arXiv:2308.07413.

---

### Official Review · Reviewer_X7ZF · 2024-11-03

**Soundness:** 2
**Presentation:** 3
**Contribution:** 2
**Rating:** 5
**Confidence:** 4

**Summary:**

This paper presents ATOMFLOW, a novel deep generative model under the flow-matching framework for the design of ligand-binding proteins from the 2D target molecular graph alone.  It co-design the protein binder structure and the target-molecule flexibility. Experimental shows the method's effectiveness.

**Strengths:**

S1. This method is based on full-atom level protein and molecule generation,  with both the protein binder and the molecule pose is designed.
S2. Experiments show the effectiveness of the method.

**Weaknesses:**

W.1 I think the major weakness lies in its novelty. First of all, using Flow-matching with full atom generation in peptide [1] or protein [2] other bio-molecules is not a new idea. Besides, the equivariant generation process [3,4,5] and FAPE loss in AlphaFold still follow previous works. Therefore,this work appears to be a fusion of several previous studies, applied in a new task, lacking innovation in terms of the methodology. I hope the author could clarify the difference between the techniques used in this paper and the origianl ones, and the  improvements upon previous approaches. This would give the authors an opportunity to highlight any methodological innovations that may not be immediately apparent. Therefore, I have doubts about whether the level of novelty in this paper meets the standards expected for a top-tier conference like ICLR.

W.2 In the experiment, RFDiffusion based method like RFDiffusionAA is compared, but there should be more works which can fulfill such tasks, such as [6] and [7]. I am wondering that can such baselines be all included into experimental comparison? I hope the author  could discuss any technical or practical limitations that may have prevented including other baselines. Besides, I think AlphaFold3 can also fulfill this task, so what is the advantage of yours over AlphaFold3? Or there are several distinct differences between AtomFlow and AlphaFold3 in design protein binder with molecule targets? I hope the authors can clarify if AlphaFold3 is directly applicable to this specific task of designing ligand-binding proteins from 2D molecular graphs, and if so, I hope the authors to explain the key differences or advantages of their approach compared to AlphaFold3 for this particular application.

[1] Kong, Full-Atom Peptide Design with Geometric Latent Diffusion, https://arxiv.org/abs/2402.13555

[2] Avishek Joey Bose, SE(3)-Stochastic Flow Matching for Protein Backbone Generation, https://arxiv.org/abs/2310.02391

[3] Xu, GeoDiff: a Geometric Diffusion Model for Molecular Conformation Generation, https://arxiv.org/abs/2203.02923

[4] Emiel, Equivariant Diffusion for Molecule Generation in 3D, https://arxiv.org/abs/2203.17003

[5] Lin, PPFlow: Target-aware Peptide Design with Torsional Flow Matching, https://arxiv.org/abs/2405.06642

[6] Zhang, Pocketgen: Generating Full-Atom Ligand-Binding Protein Pockets. bioRxiv, 2024.

[7] Hannes Stark, Bowen Jing, Regina Barzilay, and Tommi Jaakkola. Harmonic Self-Conditioned Flow Matching for joint Multi-Ligand Docking and Binding Site Design. In Forty-first International Conference on Machine Learning

**Questions:**

My major question is about the novelty and insufficient experiments. I hope the author can help to address my concerns.

---

> ### Author Response · Authors · 2024-11-19
> **A kind rebuttal to reviewer X7ZF (Part 1/3)**
>
> Thank you for your insightful feedback. We recognize your concerns regarding the novelty and experimental sufficiency of our work. In this rebuttal, we address these issues comprehensively by:
> - Demonstrating AtomFlow’s methodological contributions and its distinctions from prior works.
> - Presenting additional experimental comparisons and explaining the exclusion of certain baselines.
> - Discussing the applicability and limitations of AlphaFold 3 in our setting.
>
> ## On Methodological Novelty
> We appreciate your comments regarding the clarity of AtomFlow’s novelty. We acknowledge the relevance of the works you mentioned and have expanded our discussion to emphasize AtomFlow’s unique contributions. Below, we explicitly compare our methodology with existing literature to clarify its novelty.
> - **A new paradigm for ligand-binding protein design**
>
>     AtomFlow addresses a key challenge in real-world flexible design of ligand-binding proteins: the lack of predefined templates or fixed binding poses. Existing methods often assume a known ligand conformation [2] or binding template [3,4], which is rarely available in de novo design. This gap limits their applicability to novel targets [1]. AtomFlow overcomes this by generating both protein and ligand structures without requiring prior knowledge of binding poses, which provides a useful tool for biological discovery.
>
> - **A novel architecture for direct design of interaction**
>
>     A critical aspect of ligand binder generation lies in the fusion of two different types of biomolecules. Protein design models commonly use rigid frames for translation and rotation [5,6], whereas ligand generation methods rely on torsion angles derived from chemical graphs [7]. These distinction pose challenges for capturing spatial interactions between them. AtomFlow addresses this with a unified architecture that directly enables the modelling of biological interaction between small molecule atoms and protein residues, treating both of them as a series of movable biotokens, and defininig a probability flow on representative atomic coordinates. This unified approach also supports the generation of diverse biomolecule types, though this work primarily focuses on proteins and ligands. AF3 has tackled similar challenges, but it is (1) primarily centered on predicting interactions rather than designing interactions, and (2) it was not publicly available at the time our work was completed. This was also mentioned in your review, and we will discuss it in more detail later.
>
> - **A probability flow on conformation, not representation**
>
>     AtomFlow defines probability flow directly on molecular conformations (which could be represented by many different coordinate representations under global translation and rotation) rather than specific coordinates. This approach treats all SE(3)-transformed variants of a conformation as equivalent, ensuring that the training objective is invariant to global translation and rotation. In contrast, mainstream protein design models [5,6,7,8] rely on coordinate-based representations and train with mean squared error (MSE) loss. This penalizes structurally similar conformations under different SE(3) transformations, making the generation process less robust. By avoiding these limitations, AtomFlow offers a more natural framework for biomolecular design.
>
> - **Augmented model structure for unified features**
>
>     We repurposed key building blocks from AlphaFold 2 to construct a prediction network capable of generating structures from noisy inputs. While leveraging core components such as IPA, we enhanced the architecture to support arbitrary biomolecules by introducing atom frames and a unified biotoken embedder. This design allows for flexible conditional inputs at both sequence and token pair levels, enabling the model to accommodate diverse control requirements without further modifications. These improvements not only extend the applicability of our framework but also lay the groundwork for future extensions in biomolecular design.
>
> We've revised Sections 1 of our paper to explicitly highlight our methodological contribution.

---

> ### Author Response · Authors · 2024-11-19
> **A kind rebuttal to reviewer X7ZF (Part 2/3)**
>
> ### On Experiment Sufficiency
>
> We appreciate your suggestion regarding additional baselines such as PocketGen [3] and FlowSite [4]. While these models address ligand-protein interactions, their focus is limited to refining pocket residues within a predefined radius. They lack the capacity to design full protein folds, making direct comparison with AtomFlow infeasible. Following your suggestion, we further conducted an unfair experiment with PocketGen by providing a template binder to it, as detailed below. Despite this, the results demonstrate that AtomFlow consistently outperforms PocketGen in terms of fold quality across all radii.
>
> > For this experiment, we used the natural binders of four ligands—FAD (7bkc), SAM (7c7m), IAI (5sdv), and OQO (7v11)—as input. To evaluate the design capability of PocketGen (PG) under different constraints, we progressively increased the design radius (minimum distance to ligand) from 3.5 to 9.5. The masked target area expanded with the radius, requiring the model to redesign increasingly larger regions of the protein. When the radius exceeded the protein's dimensions (radius ≥ the protein size), all residues were masked, simulating our full-design setting. The table below presents the **min/median scRMSD** values for designs generated by PocketGen at each radius. For reference, scRMSD < 2 is generally considered a successful design. Notably, PocketGen’s performance deteriorated significantly as the radius increased, reflecting its reliance on template residues. (At radius=8 for OQO, PocketGen generated designs with several residues misaligned with the ligand, leading to abnormally high scRMSD values.) Additionally, PocketGen does not support radius settings beyond 10, preventing direct simulation of AtomFlow’s fully template-free design scenario. The results of AtomFlow is derived from our main experiment.
> |ligand|AtomFlow (r=inf)|PG (r=3.5)|PG (r=5)|PG (r=6.5)|PG (r=8)|PG (r=9.5)|
> |-|-|-|-|-|-|-|
> |FAD|**0.79/3.74**|7.10/7.38|6.75/7.81|7.29/8.35|20.92/24.23|23.12/25.23|
> |SAM|**0.83/2.01**|2.12/2.62|2.77/2.99|2.94/4.03|12.39/14.49|13.79/14.74|
> |IAI|**0.56/1.82**|0.71/0.85|0.95/1.02|2.04/2.28|3.59/5.53|9.02/11.71|
> |OQO|**0.59/1.63**|1.20/1.26|1.70/1.79|2.40/2.45|11.59/11.94|2.13/2.41|
>
> Notably, PocketGen’s performance deteriorates significantly as the radius increases, highlighting its reliance on template information. In contrast, AtomFlow maintains high performance without such dependencies. Based on similar settings, we expect FlowSite to exhibit comparable performance patterns to PocketGen.
> We added explicit description on the exclusion of these methods in our paper.

---

> ### Author Response · Authors · 2024-11-19
> **A kind rebuttal to reviewer X7ZF (Part 3/3)**
>
> ### On AlphaFold 3
> Regarding AlphaFold 3, while its diffusion-based approach represents an advance in biomolecule prediction, its heavy reliance on predefined sequences limits its applicability to de novo protein-ligand design. In our experiments, neither AlphaFold 3 nor its replications (e.g., Protenix [9], Chai [10]) could reliably generate ligand-binding proteins from scratch. This highlights AtomFlow’s tailored capabilities for this task.
>
> - While AlphaFold 3 does not directly apply to our setting, we found your suggestion of adapting a structure prediction model for design tasks to be highly insightful. Inspired by this, we conducted a preliminary experiment using two AlphaFold 3 replications, Protenix and Chai, which were forcibly adapted to accept a series of [UNK] residues along with a ligand (FAD) as input. Our findings are as follows: approximately 20% of proteins generated by Protenix achieved scRMSD < 2, with reasonable ligand conformations but no visible interactions between ligand and protein. Chai performed slightly better, with around 50% of its generated proteins achieving scRMSD < 2, though none of the ligands exhibited correct conformations. Notably, the original AlphaFold 3 model is hard-coded to require a valid residue sequence, rendering direct comparisons infeasible. These results suggest that while AlphaFold 3 shows potential, it currently cannot generalize to binder design without substantial retraining and engineering, which falls outside the scope of this work.
>
> - At the time of writing our draft, no publicly available AlphaFold 3 model was available for local experimentation. Inspired by your suggestion, we revisited AlphaFold 3 and leveraged recent replications (e.g., Chai) to conduct a new experiment. Specifically, we developed an alternative in silico metric for binding affinity based on minimum interchain pAE predictions, inspired by AlphaProteo [11]. Using this metric, AtomFlow demonstrated superior results compared to RFDiffusionAA. This new experiment has been added to the appendix, and we hope that introducing this metric to the community will further support our contribution.
> - Additionally, we hypothesize that AlphaFold 3 could serve as a foundation for protein hallucination approaches, similar to BindCraft [12], which utilizes AlphaFold 2. Although this direction lies beyond the scope of our current work, it represents a promising avenue for future exploration.
>
> We hope our explanation has addressed your questions and concerns thoroughly. We sincerely appreciate your valuable feedback and are always happy to clarify further or discuss ways to improve our work. If you find our responses satisfactory, we would be grateful for your consideration of a higher score. Thank you!
>
> [1] Bick, M. J., Greisen, P. J., Morey, K. J., Antunes, M. S., La, D., Sankaran, B., ... & Baker, D. (2017). Computational design of environmental sensors for the potent opioid fentanyl. Elife, 6, e28909.
>
> [2] Krishna, R., Wang, J., Ahern, W., Sturmfels, P., Venkatesh, P., Kalvet, I., ... & Baker, D. (2024). Generalized biomolecular modeling and design with RoseTTAFold All-Atom. Science, 384(6693), eadl2528.
>
> [3] Zhang, Z., Shen, W. X., Liu, Q., & Zitnik, M. (2024). Efficient generation of protein pockets with PocketGen. Nature Machine Intelligence, 1-14.
>
> [4] Stärk, H., Jing, B., Barzilay, R., & Jaakkola, T. (2023). Harmonic self-conditioned flow matching for multi-ligand docking and binding site design. arXiv preprint arXiv:2310.05764.
>
> [5] Yim, J., Trippe, B. L., De Bortoli, V., Mathieu, E., Doucet, A., Barzilay, R., & Jaakkola, T. (2023). SE (3) diffusion model with application to protein backbone generation. arXiv preprint arXiv:2302.02277.
>
> [6] Yim, J., Campbell, A., Foong, A. Y., Gastegger, M., Jiménez-Luna, J., Lewis, S., ... & Noé, F. (2023). Fast protein backbone generation with SE (3) flow matching. arXiv preprint arXiv:2310.05297.
>
> [7] Lin, Y., Lee, M., Zhang, Z., & AlQuraishi, M. (2024). Out of Many, One: Designing and Scaffolding Proteins at the Scale of the Structural Universe with Genie 2. arXiv preprint arXiv:2405.15489.
>
> [8] Abramson, J., Adler, J., Dunger, J., Evans, R., Green, T., Pritzel, A., ... & Jumper, J. M. (2024). Accurate structure prediction of biomolecular interactions with AlphaFold 3. Nature, 1-3.
>
> [9] https://github.com/bytedance/Protenix
>
> [10] Chai Discovery, Boitreaud, J., Dent, J., McPartlon, M., Meier, J., Reis, V., ... & Wu, K. (2024). Chai-1: Decoding the molecular interactions of life. bioRxiv, 2024-10.
>
> [11] Zambaldi, V., La, D., Chu, A. E., Patani, H., Danson, A. E., Kwan, T. O., ... & Wang, J. (2024). De novo design of high-affinity protein binders with AlphaProteo. arXiv preprint arXiv:2409.08022.
>
> [12] Pacesa, M., Nickel, L., Schmidt, J., Pyatova, E., Schellhaas, C., Kissling, L., ... & Correia, B. E. (2024). BindCraft: one-shot design of functional protein binders. bioRxiv, 2024-09.

---

> ### Comment · Reviewer_X7ZF · 2024-11-19
> **Thanks a lot for your response.**
>
> We thank a lot for the detailed response of the authors. The extra expriments solve a lots of my concerns. Besides, the detailed explanation of the methods compared with other competitors is convincing. However, I still think that the contribution of method novelty is limited, since the use of probabilistic models has already been thoroughly explored in the relevant field, and the application of flow matching in molecular and protein design has become quite common.
>
> Therefore, after reviewing your response, I reread the article and reassessed its quality. I have decided to raise my initial rating from 3 to 5.

---

> > ### Author Response · Authors · 2024-11-20
> > **Thank You for Your Feedback and Updated Rating**
> >
> > Thank you for your thoughtful feedback and for reassessing our manuscript. We are pleased that the additional experiments and detailed explanations have addressed many of your concerns.
> >
> > While we fully understand the reviewer’s perspective on the technical novelty of our paper, we believe our method offers innovative applications in ligand-binding protein design. Our model is designed to learn binding patterns across diverse biomolecules and generate complete binders from scratch (which most of previous work fail to do), reducing reliance on prior knowledge and enhancing our ability to tackle novel targets without bound conformers. We guess we can keep both views regarding the novelty of our work.
> >
> > We sincerely appreciate your constructive comments and your decision to raise the rating. Your insights have been invaluable in improving our work.

---

### Author Response · Authors · 2024-11-19
**A Heartfelt Thanks to Our Reviewers**

We sincerely thank all reviewers for their insightful comments and valuable feedback on our paper. Below, we summarize the updates made in response to your suggestions:

### New Experiments and Metrics

Recognizing the absence of a well-defined in silico metric for the binder design problem, we initially followed the conventions of existing literature in our experiments. After engaging with the reviewers' thoughtful feedback, we introduced several new metrics to strengthen the connection between our machine learning methodology and the biological community.

During the review process, several AlphaFold 3-like models were released. Leveraging this advancement, we added a new in silico experiment in the appendix to evaluate binding affinity, using the minimum interchain pAE predicted by AlphaFold 3 as a proxy for affinity, inspired by the approach in AlphaProteo.

We also added metrics to assess the geometric distribution and chemical validity of AtomFlow’s generated results. Additionally, we included diversity and novelty results for RFDiffusion-AA in the appendix.

### Supporting Results

To explicitly demonstrate that pocket design models are not suitable for the full protein binder generation setting, we conducted and reported experiments on PocketFlow in the appendix. We also performed a speed comparison between AtomFlow and RFDiffusion-AA, showing that AtomFlow is approximately 5× faster. This justifies our initial decision not to provide baseline results in the supplementary extended experiments (we plan to include these baseline results in the final version). The speed test further highlights the added value of AtomFlow.

### Text Refinement

We refined the text in Section 1 to better explain our contribution. We added an explicit introduction of excluding pocket design models in Section 5. We also added more details of the loss and our training process in the appendix.

We addressed each reviewer's concern in our reply to the review. We sincerely appreciate your feedback and are always happy to clarify further or discuss ways to improve our work.

---

### Meta-Review · Area_Chair_B9Hx · 2024-12-22

**Metareview:**

The paper presents a method, AtomFlow, to design protein structures that bind to (a small molecule) input ligands. Contrary to prior work, it does so without the assumption that the ligand’s bound pose is given by adopting a flow that jointly generates the structure of the protein and the binding ligand. The general method is a parallel to the diffusion-based RFDiffAA that is based on flow matching. It evaluates the method against RFDiffAA baselines and shows comparable designability results, without knowing the bound structure, both on the RFDiffAA benchmark as well as on a new set of ligands.

The reviewers acknowledged the importance of the task, the suitability of the architecture and novel use of flow matching, the clarity of the writing and presentation, the meaningful results in the experiments, the practical usefulness of the method,

On the other hand, the reviewers were concerned about the lack of technical novelty in light of prior work such as AlphaFlow and RFDiffAA (for a technical paper), and –as an application paper– they were concerned about  proper and conclusive comparison with baselines, the size of the test set, the extent and quality of evaluation tasks and metrics.

The authors did a thorough rebuttal which addressed some of the concerns and provided additional results and insights, through which the paper clearly improved as indicated by the reviewers. Some concerns were fully resolved such as comparison with methods beyond RF-DIFF family and several clarifications on how the evaluation is currently done.

The AC believes the current paper has several merits and majorly improved during the rebuttal period as indicated by the reviewers. However, during the discussion among the reviewers and after the rebuttal period, the reviewers felt that the paper can further improve along the directions mentioned in the original reviews and during the rebuttal. The AC agrees with this evaluation and believes it is important that, after the suggested improvements, the paper goes through another round of review. Therefore, the AC suggests rejection at this time. A resubmission considering all the valuable feedback from the reviewers should make the paper much stronger for the next top venue. The main points that can improve the paper are the suggested directions to extend the empirical evidence so that it becomes a stronger application paper. Alternatively, or in tandem, the paper can better highlight what technical contribution the paper has that goes beyond the application of the paper.

**Additional Comments On Reviewer Discussion:**

Four expert reviewers reviewed the paper covering all the main aspects of the work from the application to the technical aspects. They all eventually rated the paper at the borderline and therefore a thorough discussion happened with the authors and continued in the next phase. The reviewers and AC acknowledged the merits of the submission, and follow-up improvements but yet led to a reject decision as, for an application paper, the empirical evidence can become stronger and more extensive.

---

### Decision · Program_Chairs · 2025-01-22

Reject